



# Regulation of carbon dioxide and methane in small agricultural reservoirs: Optimizing potential for greenhouse gas uptake

Jackie R. Webb[1]*, Peter R. Leavitt[1,2,3], Gavin L. Simpson[1,2], Helen Baulch[4], Heather A. Haig[1], Kyle R. Hodder[5], Kerri Finlay[1]

[1]Department of Biology, University of Regina, Regina, SK, S4S0A2, Canada.
[2]Institute of Environmental Change and Society, University of Regina, Regina, Saskatchewan, Canada, S4S 0A2
[3]Institute for Global Food Security, Queen's University Belfast, Belfast, Northern Ireland, BT7 1NN, United Kingdom.
[4]School of Environment and Sustainability, Global Institute for Water Security, University of Saskatchewan, 11 Innovation Boulevard, Saskatoon, SK S7N3H5, Canada
[5]Department of Geography & Environmental Studies, University of Regina, Regina, SK, S4S0A2, Canada.

*Correspondence to:* Jackie R. Webb (jackie.roslyn.webb@gmail.com)

**Abstract.** Small farm reservoirs are abundant in many agricultural regions across the globe and have the potential to be large contributing sources of carbon dioxide ($CO_2$) and methane ($CH_4$) to agricultural landscapes. Compared to natural ponds, these artificial waterbodies remain overlooked in both agricultural greenhouse gas (GHG) inventories and inland water global carbon (C) budgets. Improved understanding of the environmental controls of C emissions from farm reservoirs is required to address and manage their potential importance. Here, we conducted a regional scale survey (~235,000 km$^2$) to measure $CO_2$ and $CH_4$ concentrations and diffusive fluxes across 101 small farm reservoirs in Canada's largest agricultural area. A combination of abiotic, biotic, hydromorphologic, and landscape variables were modelled using generalized additive models (GAMs) to identify regulatory mechanisms. We found that $CO_2$ concentration was best estimated by a combination of internal metabolism and groundwater-derived alkalinity (65.7% deviance explained), while multiple lines of evidence support a positive association between eutrophication and $CH_4$ production (74.1% deviance explained). Fluxes ranged from -21 to 466 and 0.14 to 92 mmol m$^{-2}$ d$^{-1}$ for $CO_2$ and $CH_4$, respectively, with $CH_4$ contributing an average of 74% of $CO_2$-equivalent ($CO_2$-e) emissions. Approximately 19% farm reservoirs were found to be net $CO_2$-e sinks. From our models, we show that the GHG impact of farm reservoirs can be greatly minimised through overall improvements in water quality and the construction and maintenance of deeper reservoirs.



## 1 Introduction

Small waterbodies have recently been recognised as substantial contributors to global carbon emissions from inland waters.
Current assessments estimate that small ponds ($<0.001$ km$^2$) account for 15% and 40% of global $CO_2$ and $CH_4$ emissions respectively from lakes (Holgerson and Raymond, 2016). Other estimates suggest emissions from small lakes and impoundments (0.001 to 0.01 km$^2$) could constitute 40% of global $CO_2$ emissions and 20% of global $CH_4$ emissions from lentic ecosystems (DelSontro et al., 2018). Extreme $CO_2$ and $CH_4$ supersaturation is characteristic of small waterbodies due to greater contact with the sediment and littoral zone (Downing et al., 2008; Holgerson, 2015), often making them
disproportionately important in landscape carbon (C) budgets (Hamilton et al., 1994; Premke et al., 2016; Kuhn et al., 2018). Conversely, ponds may have the capacity to store landscape-significant amounts of carbon, with burial rates 20–30 times higher than wetlands and large lakes (Gilbert et al., 2014; Taylor et al., 2019). While these assessments have stimulated a growing area of research on small waterbodies, much work is still needed to revise estimates of their carbon emissions due to limited knowledge on their regional distribution and variability, as well as their overall global extent (Verpoorter et al.,
2014). This is particularly true for greenhouse gas (GHG) emissions from human-created small waterbodies.

The expansion of agriculture and urban land use has introduced a new type of lentic system that remains relatively unexplored – small artificial waterbodies (Clifford and Heffernan, 2018). These artificial aquatic systems have been created through human modification of the hydrological landscape and include small farm reservoirs and urban ponds. Farm reservoirs are earthen excavations designed to store water for later use (BC Ministry of Agriculture, 2013). The global
abundance of these systems remains uncertain (Verpoorter et al., 2014), but statistical extrapolation suggest there may be around 16 million worldwide (Lehner et al., 2011). Regional-scale inventories indicate that upwards of 8 million farm reservoirs exist in the USA (Brunson, 1999; Smith et al., 2002), China (Chen et al., 2019), India (Anbumozhi et al., 2001), South Africa (Mantel et al., 2017), and Australia alone (Lowe et al., 2005; MDBA, 2008; Grinham et al., 2018a). The density of farm reservoirs can exceed 30% of agricultural area in some regions such as China where food demand is high
(Chen et al., 2019). Small agricultural reservoirs are estimated to cover 77,000 km$^2$ globally and are being created at rates up to 60% of standing stock per annum in some regions (Downing et al., 2008). Given their abundance, these artificial systems may contribute substantially to landscape biogeochemical cycles, including fluxes of GHG. In particular, very little is known of the capability of these systems to act as GHG sinks to partially offset the otherwise strong carbon efflux associated with intensive agriculture (Robertson et al., 2000).

Understanding the controls and rates of carbon fluxes from small artificial waterbodies is the first step required to understand their landscape and eventually global importance. Farm reservoirs have the potential to be potent sources of $CO_2$ and $CH_4$ due to their highly eutrophic nature (Downing et al., 2008) and high degree of heterotrophy (Holgerson and Raymond, 2016). Further, estimates of $CO_2$ and $CH_4$ flux are complicated by high variation among reservoirs and regions in the importance of groundwater, littoral macrophytes, and local land use practises (Pennock et al., 2010; Badiou et al., 2019).
Currently, only three studies have comprehensively assessed C fluxes from small agricultural reservoirs and these support





the notion that they are important landscape sources of GHGs (Panneer Selvam et al., 2014; Grinham et al., 2018a; Ollivier et al., 2019). All studies found large fractions of $CH_4$ being released, and large mean $CO_2$ emissions on the order of 24 and 99 mmol m$^{-2}$ d$^{-1}$, comparable to the global average flux rate of very small natural ponds (35 mmol m$^{-2}$ d$^{-1}$, Holgerson and Raymond, 2016). However, carbon fluxes from farm reservoirs remain unaccounted in agricultural GHG inventories and

global inland water carbon budgets. To facilitate their inclusion in agricultural and global budgets, we need to further constrain flux rates and mechanisms across a broad geographic area.

Here, we present a large-scale assessment of $CO_2$ and $CH_4$ fluxes from small farm reservoirs in the Northern Great Plains, the largest agricultural region in Canada. The hydroclimate, lithology and edaphic features are vastly different compared to previous studies of agricultural areas (Australia, India, USA), with factors that favour $CO_2$ uptake by alkaline surface waters

(Finlay et al., 2009; Finlay et al., 2015) and lead to high variability in $CH_4$ fluxes from regional wetlands (Pennock et al., 2010; Badiou et al., 2019). Our aim was to identify the key environmental conditions regulating $CO_2$ and $CH_4$ fluxes, and to evaluate the potential for reservoir design to minimize $CO_2$-equivalent ($CO_2$-e) emissions. To achieve this goal, we carried out an extensive survey of $CO_2$ and $CH_4$ concentrations across 101 farm reservoirs and used generalized additive models (GAMs) to assess the effects of abiotic, biotic, hydromorphological and land use properties. Our findings show that farm

dams were not always strong sources of carbon emissions and in certain cases can be carbon neutral or sinks in terms of $CO_2$-e emissions. By identifying the driving characteristics of farm dams that support reduced C emissions, our findings provide the first step to developing management strategies to help minimise farm carbon emissions.

## 2 Methods

### 2.1 Study site

Farm sites were surveyed across the agricultural region of Saskatchewan, Canada (Fig. 1). This region covers an area of 235,000 km$^2$ in the southern half of the province, where agriculture accounts for ~80% of land use. The region has a sub-humid to semi-arid climate (Köppen D*fb* classification), with short warm summers (~18°C) and long winters (~-17°C) resulting in 4.5 to 5.5 months of ice cover on surface waters (Finlay et al., 2015).

Small farm reservoirs (known locally as 'dugouts') are a prominent feature of the landscape, with densities up to 10 per km$^2$

(Fig. 1B). Up until 1985, over 110,000 farm reservoirs had been constructed in Saskatchewan (Gan, 2000), although subsequent densities are unknown. We sampled 101 farm reservoirs between July and August 2017, ranging in surface area from 158 – 13,900 m$^2$ (Table 1), including basins in pasture (n = 80) and cropland (n = 21) sites. Saskatchewan farm reservoirs are typically uniform in shape and morphometry, dug to a depth of 4 to 6 m with steep sides (1.5:1 slopes). Most shallow wetlands and lakes in the region exhibit water balances dominated by evaporation and limited inflow from winter

precipitation or groundwater (Conly and van der Kamp, 2001; Pham et al., 2009). Farm reservoirs differ from small natural waterbodies in that they have a higher ratio of water volume to surface area, designed to minimise evaporation losses. Despite this feature, arid conditions persisted during the sampling year, with reduced (34-65%) annual rainfall such that



many reservoirs were only half their designed depth. Natural waterbodies also tend to be high pH hard-water systems, owing to the soils which consist of glacial till high in carbonates (Last and Ginn, 2005). The same was observed for the majority of
farm reservoirs, with an average pH of 8.75 (Table 1).

## 2.2 CO₂ and CH₄ measurements

Dissolved gas samples were collected using the in-field headspace extraction method (Webb et al., 2019). Briefly, water was collected from ~30 cm below the surface using a submersible pump which filled a 1.2-L glass-serum bottle, ensuring the bottle overflowed and no air bubbles were present. The bottle was sealed with a rubber stopper fitted with two three-way
stopcock valves. Using two 60-mL air-tight syringes, atmospheric air was added to the bottle whilst simultaneously extracting 60-mL of water. The bottle was then shaken for 2 minutes to ensure gas equilibration in the headspace. Two analytical replicates were extracted and stored in 12-mL evacuated Exetainer vials with double-wadded caps. Headspace concentrations of $CO_2$ and $CH_4$ were measured using gas chromatography with a Scion 456 Gas Chromatograph (Bruker Ltd.) and calculated using standard curves. Dry molar fractions were corrected for dilution and converted to concentrations
according to solubility coefficients (Weiss, 1974; Yamamoto et al., 1976).

Carbon dioxide and methane fluxes were estimated for each water body using water column concentrations ($C_{water}$) and average farm reservoir gas transfer velocity ($k_c$) using the following equation:

$$f_C = k_c(C_{water} - C_{air}), \tag{1}$$

where $f_c$ is the flux of $CO_2$ or $CH_4$ (mmol m$^{-2}$ d$^{-1}$) and $C_{air}$ is the ambient air concentration. The average global mixing ratios
for the sampling period of 406 and 1.85 µatm were used for ambient concentrations for $CO_2$ and $CH_4$ respectively (Mauna Loa NOAA station, June to August 2017). Site-specific gas transfer velocity ($k_c$) was determined from 30 individual floating- chamber (area = 0.23 m$^2$, volume = 0.046 m$^3$) measurements carried out on a subset of 10 reservoirs. During each 10-minute incubation, changes in gas concentrations were measured at 2.5-min intervals by taking samples using syringes and dispensing gases into pre-evacuated 12-mL vials. The flux (mmol m$^{-2}$ d$^{-1}$) was calculated from the observed rate of
change in the dry mole fraction of the respective gas (Lorke et al., 2015). The gas transfer velocity normalised to a Schmidt number of 600 ($k_{600}$) for each respective gas was then determined using measured flux, *in situ* gas concentrations, atmospheric concentration, Henry's constant, and Schmidt numbers, assuming a Schmidt exponent of 0.67. The average $k_{600}$ calculated from the floating chamber incubations was 1.50 ± 1.34 m d$^{-1}$ and 1.68 ± 1.26 m d$^{-1}$ for $CO_2$ and $CH_4$, respectively.

## 2.3 Abiotic and biotic variables

A range of abiotic and biotic parameters were measured at each site. Water quality variables including temperature (°C), pH, dissolved $O_2$ (DO; mg $O_2$ L$^{-1}$), conductivity (µS cm$^{-2}$), and salinity were measured at 0.5-m intervals from the surface to the bottom using a YSI (Yellow Springs Instruments, OH, USA) multi-probe meter. Surface (0.5 m) samples for water chemistry were collected using a submersible pump. Upon collection, samples for dissolved nitrogen ($NO_3+NO_2$, $NH_4$, total





dissolved N; μg N L$^{-1}$), soluble reactive phosphorus (SRP; μg P L$^{-1}$) and total dissolved P (TDP; μg P L$^{-1}$), dissolved organic

and inorganic carbon (DOC, DIC; mg C L$^{-1}$), alkalinity (OH + HCO$_3$ + CO$_3$; mg L$^{-1}$ as CaCO$_3$), and water isotopes ($\delta^2$H, $\delta^{18}$O; ‰) were filtered through a 0.45-μm pore membrane filter. Nutrient and dissolved carbon samples were stored in a dark bottle at 4°C until analysis. Chlorophyll *a* (Chl-*a*) samples were collected on GF/C glass-fiber filters (nominal pore size 1.2 μm) and frozen (-10°C) until analysis. Sediment samples were collected at the centre of each reservoir, the uppermost 10 cm using an Ekman grab sampler, and were frozen at -10°C until analysis.

Most analyses were carried out at the University of Regina Institute of Environmental Change and Society (IECS). Water nutrient and dissolved carbon concentrations were measured on a Lachat QuikChem 8500 and Shimadzu model 5000A total carbon analyzer, following standard analytical procedures, respectively (Patoine et al., 2006;Finlay et al., 2009). Alkalinity was measured using standard methods of the US Environmental Protection Agency (EPA) on a SmartChem 200 Discrete Analyser (WestCo) and estimated as the concentration of CaCO$_3$ (EPA, 1974). Chl-*a* was analysed using standard

trichromatic methods (Finlay et al. 2009). The total carbon and nitrogen content (% dry weight) of freeze-dried sediment samples were determined on a NC2500 Elemental Analyzer (ThermoQuest, CE Instruments).

## 2.4 Hydromorphology

Morphometric parameters of reservoirs were estimated for each site. The depth of each farm reservoir was measured during using a portable ultrasonic depth sounder, taken at the deepest section in the centre of the reservoir. Surface area was

determined using Google Earth satellite imagery. Reservoir volume was calculated using the formula for a prismoid by assuming that all sites maintained their original shape, including slopes of 1.5:1 ratio (Andresen et al., 2015). From these measurements, an Index of Basin Permanence (IBP) was calculated (Kerekes, 1977).

The degree of water-column mixing or vertical stratification was determined by calculating the squared Brunt-Väisälä buoyancy frequency ($N^2$, s$^{-2}$). The strongest density gradient was calculated based on vertical temperature measurements at

0.5-m depth intervals using the package *rLakeAnalyzer* (Read et al., 2012) in R (version 3.5.2; R Core Team 2018).

The hydrology of farm reservoirs was estimated through analysis of $\delta^{18}$O and $\delta^2$H isotope values of water. Samples were collected from 0.5 m below the surface, filtered (0.45-μm pore) and stored in amber borosilicate jars at 4°C until analysis using a Picarro L2120-I cavity ring-down spectrometer (CRDS). Hydrological parameters, including evaporation to inflow ratio (E/I), residence time (years), and inflow volume (m$^3$), deuterium ($^2$H) excess (d-excess), and $\delta^{18}$O inflow ($\delta_I$) values,

were calculated using the coupled isotope tracer method (Yi et al., 2008) and conventional isotopic water-balance methods (Gibson et al., 2001). All methods assumed that reservoirs were headwater systems in hydrological steady-state (Yi et al., 2008). Model inputs included information about the local water meteoric line (LWML), the trajectory of evaporation along a local evaporative line (LEL), and regional meteorological conditions. From here, the water mass balance of a given waterbody can be quantified based on its relative position along the LEL (Gibson et al., 2001).



Briefly, the isotopic inflow values were estimated by the intercept between the LWML and site-specific LEL as determined by $\delta^{18}O$ evaporation value ($\delta_E$) and $\delta^{18}O$ reservoir water value at each site (Yi et al., 2008). The E/I ratio was calculated by using headwater isotopic models of the water mass balance $((\delta_I - \delta_L) * (\delta_E - \delta_L)^{-1})$. Hydrologic residence time was estimated from the reservoir volume and the water isotopic values of waterbodies, inflow, and evaporation. Deuterium excess (d-excess ‰ = $\delta^2H - 8*\delta^{18}O$) was calculated as an additional indicator of evaporation losses, where lower values (< -10‰)

indicate isotopic enrichment from precipitation (Brooks et al., 2014).

## 2.5 Landscape properties

Landscape soil data was obtained from The National Soil DataBase, Government of Canada (http://sis.agr.gc.ca/cansis/nsdb/dss/v3/index.html) using ArcGIS to extract the soil attributes at each site. Extracted variables included soil salinity, soil pH, soil organic carbon content, saturated hydraulic conductivity ($K_{sat}$), cation exchange capacity

(CEC), and the total composition of soil from sand, silt, and clay fractions (%). Reservoir elevation (m, a.s.l.) was determined using ArcGIS and the Canadian Digital Elevation Model (CDEM, v1.1). Local land use in the immediate area surrounding each reservoir was categorised into three types based on local observations at the time of sampling. Categories included pasture land used for either livestock grazing or hay harvesting, pasture where livestock have direct access to the waterbody, and crop fields.

## 2.6 Statistical analyses

Environmental variables were selected based on known or presumed influence on $CO_2$ and $CH_4$ concentrations in lakes and small waterbodies. Both biotic and abiotic predictors that influence production or consumption of $CO_2$ and $CH_4$ were selected, including DO, alkalinity, $NO_X$, $NH_4$, dissolved inorganic nitrogen (DIN), TDN, TDP, Chl-*a*, DOC, conductivity, pH, and sediment organic C:N ratio. The influence of reservoir hydrology and morphology were also examined, including

measures of surface area, basin permanence, hydrologic regime (E/I), water source ($\delta_I$), and degree of mixing (or stratification). Finally, potential effects of the surrounding terrestrial landscape were estimated in models using soil properties, elevation, and land use practises to account for any localised landscape drivers. Before testing relationships, all predictors were transformed as needed using either $\log_{10}$ or square root to remove skewness.

The relationships between covariates and $CO_2$ and $CH_4$ were estimated using generalised additive models (GAMs). GAMs

provide an ideal approach to model non-linear associations between predictor variables and responses, using the sum of unspecified smooth functions to estimate trends. GAMs were developed with a gamma distribution for the response and the log link function. Each model included covariates that represented hydromorphological, abiotic and biotic, and landscape controls. To avoid multicollinearity, correlation coefficients between pairs from Pearson linear correlation tests was used to guide covariate choice before model fitting (Table S1-3). Candidate variables were then selected for each model to test

which variables best estimate variability in $CO_2$ and $CH_4$ concentrations. All model coefficients were estimated using





restricted marginal likelihood with the *mgcv* package (Wood, 2011; Wood et al., 2016) for R (version 3.5.2; R Core Team 2018).

## 3 Results

The region experienced a drier than average year during sampling, with recorded average annual precipitation ~60% less
than the long-term climate average of 390 mm in Regina, Saskatchewan (Government of Canada, http://climate.weather.gc.ca). Consequently, while most farm reservoirs were constructed to ~5 m depth the mean water-column depth was 2.1 m (0.2-5.1, Table 1). Despite this, isotopic analysis of water revealed that 93% of waterbodies exhibited an E/I < 1.0, suggesting that reservoirs were gaining more water than was lost via evaporation. In general, water residence time was ~8 months, although the range in this value was large (29 days to 2.5 years). Estimates of inflow $\delta^{18}O$ ($\delta_I$)
indicated variable water sources, with 79% derived from rain (>-15.66‰), 6% from snowmelt or groundwater (<-17.9‰), and 15% intermediate between sources (-17.9 to -15.6‰).

Carbon dioxide and methane concentrations spanned three orders of magnitude across surveyed reservoirs, with concentrations ranging between 1.3 to 326.1 and 0.1 to 54.5 µM for $CO_2$ and $CH_4$, respectively (Table 1). Most waterbodies were alkaline, with a mean pH of 8.8 (7.0 to10.2) and carbonate alkalinity between 71 and 755 mg $L^{-1}$. Many waters were
highly eutrophic, with means for Chl-*a* of 99 µg $L^{-1}$ (range 2 to 344 µg $L^{-1}$), total nitrogen of >3,000 µg N $L^{-1}$ (418 to 14,280), and total phosphorus of 285 µg P $L^{-1}$ (9 to 648). Dissolved $O_2$ in the surface layer varied by three orders of magnitude among basins with 32% exhibiting oversaturation (>100%).

### 3.1 Models

Regional variation in $CO_2$ concentrations were best estimated in a GAM including pH alone, with 86.3% of deviance
explained and a strongly declining $CO_2$ at pH above 8 (Fig. S1). Exclusive of the model with pH, the detailed mechanistic GAM for estimating $CO_2$ concentrations across farm reservoirs included a combination of DO saturation, alkalinity, $NO_x$, thermal stratification (buoyancy frequency), basin hydrology (the interaction between $\delta_I$ and WRT), and landscape features (soil CEC, elevation, soil salinity) (Fig. 2). Overall, the model explained 65.7% of deviance in $CO_2$ concentrations (Table S4). All covariates had a significant effect except soil salinity, with DO, alkalinity, and the interaction between $\delta_I$ and WRT
being the strongest predictors ($p <0.001$). $CO_2$ concentrations displayed a positive response with increasing alkalinity, $NO_x$, buoyancy frequency, and soil CEC, with a generally negative response to increasing DO and elevation. The effect of DO on $CO_2$ was particularly distinct between 25 and 100% $O_2$ saturation (Fig. 2A). The interactive effect of hydrology parameters suggests that sites with elevated rain inflows ($\delta^{18}O > -12.5$‰) and longer WRT will exhibit undersaturated $CO_2$ concentrations.
Variation in $CH_4$ concentrations among waterbodies were explained by a combination of DO saturation, sediment C/N ratio, DIN, conductivity, the interaction between $\delta_I$ and WRT, and local land use (Fig. 3), with buoyancy frequency, soil $K_{sat}$, and





elevation not significant. Overall, the GAM explained 74.1% of the deviance in $CH_4$ (Table S5). Concentrations of $CH_4$ increased with sediment C/N and DIN and decreased with conductivity. The significant unimodal relationship with DO indicates that the highest observed $CH_4$ concentrations occurred under both anoxic and supersaturated $O_2$ environments (Fig.

3A), while low $CH_4$ levels were seen when inflow was more composed of snowmelt or groundwater (depleted isotope values) and WRT was long (Fig. 3F). In contrast to the $CO_2$ model, soil properties and elevation were not significant drivers, yet local land use was significant, with crop sites having significantly higher $CH_4$ compared to pastures.

## 4 Discussion

Our comprehensive spatial analysis revealed wide variations among $CO_2$ and $CH_4$ concentrations between farm reservoirs.

Significant modelled environmental drivers suggested $CO_2$ was primarily controlled by pH, with strong independent models indicating mechanisms associated with primary productivity, the hydrological regime, and landscape elevation. In contrast, $CH_4$ was most correlated by internal abiotic and biotic mechanisms. We discuss these potential drivers in detail and from our evidence suggest management strategies that may help reduce the net GHG effect of these farm reservoirs.

### 4.1 Environmental drivers of $CO_2$ concentrations

As seen in other hardwater ecosystems, variations in $CO_2$ were strongly coupled to differences among sites in water-column pH (Finlay et al., 2015; Müller et al., 2016). We demonstrate this with the strong correlation observed between $CO_2$ and pH in a separate GAM of only water pH as a covariate, explaining 86.3% of deviance (Fig. S1). As expected, the role of pH in regulating $CO_2$ content is most pronounced at values between 8.6-9.0, the transition point where the predominant species of DIC shifts from free $CO_2$ to $HCO_3^-$ (Duarte et al., 2008; Finlay et al., 2015). Above this value, carbonate buffering

increasingly regulates pH and restricts $CO_2$ to only trace fractions of total DIC (Stumm and Morgan 1970). However, direct changes in $CO_2$ concentrations can also alter water-column pH, such as biological metabolism (Talling, 2010). Therefore, given the direct chemical relationship between pH and $CO_2$ concentrations (Stumm and Morgan, 1970), we opted to leave pH out of our model to further investigate the underlying biological, chemical, hydrological, and land use mechanisms.

The detailed GAM showed that variance in $CO_2$ concentrations among farm reservoirs was estimated (65.7% of deviance) by

a combination of predictors related to water-column productivity and microbial metabolism (DO saturation, alkalinity, $NO_x$), thermal stratification (buoyancy frequency), basin hydrology (the interaction between $\delta_I$ and WRT), and landscape features (soil CEC, elevation) (Fig. 2), but not local soil salinity. This was shown by the DO, alkalinity, $\delta_I$ and WRT covariates having the most significant effect at $p<0.001$, while $CO_2$ concentrations did not vary significantly between different soil salinity levels (Table S4, Fig. 2).

Carbon dioxide and dissolved oxygen are closely linked by biological metabolism in aquatic systems and diverge when other chemical or physical processes occur. Here, we see evidence for both linked and divergence (Fig. 2A). The tight linear relationship between $CO_2$ and $O_2$ at 25 to 100% saturation indicates close coupling between the gases. This likely represents





control via metabolic processes such as net ecosystem production (NEP) or chemical oxidation of reduced species (Stets et al., 2017). In contrast, relationships between $CO_2$ and $O_2$ were less well defined a both high and low oxygen saturations,

conditions which may indicate a greater contribution from anaerobic production of $CO_2$ (Torgersen and Branco, 2008; Holgerson, 2015). Alternatively, alkalinity buffering can mediate the effect of NEP on $CO_2$ concentrations at both extreme ranges of the DO spectrum (Marcé et al., 2015). Alkalinity buffering is most likely to affect $CO_2$-DO relationships in waters where alkalinity is >2000 µeq $L^{-1}$ (Stets et al., 2017) which was the case for ~90% of our sites (Table 1; Fig. 2).

Stratification can also weaken the impact of DO as a driver for $CO_2$ by regulating the effect of sediment respiration on

epilimnetic chemistry (Huotari et al., 2009; Holgerson, 2015). Our model shows that those sites that were most stratified (elevated buoyancy frequency) exhibited higher $CO_2$ concentrations (Fig. 2D). This pattern contrasts those observed in other small lentic systems where elevated epilimnetic $CO_2$ concentrations were observed during and after breakdown of water-column stratification (Huotari et al., 2009; Glaz et al., 2016). Preliminary seasonal studies of some farm reservoirs in 2018 show that stratification is strong and persistent throughout the summer, with no obvious diurnal mixing events. Such strong

stratification can maintain anoxic conditions throughout most of the water column, which supports intense anaerobic respiration and $CO_2$ production. Here, the higher $CO_2$ concentrations observed at high stratification may simply represent an accumulation of benthic-derived respiratory $CO_2$ during summer that eventually diffuses into the epilimnion.

The positive association between $NO_x$ and $CO_2$ found in our reservoirs is consistent with similar patterns seen with dissolved inorganic N species in other artificial waterbodies (Ollivier et al., 2019; Peacock et al., 2019) and regional prairie lakes

(Wiik et al., 2018). In some lakes, high N loading favoured elevated heterotrophy, despite simultaneous boosts in primary production which draws down free $CO_2$ (Huttunen et al., 2003; Cole et al., 2000). The effect of a high N influx on $CO_2$ may be heightened in smaller or shallow lentic waters which are more influenced by sedimentary processes (Torgersen and Branco, 2008). Further, high N availability can increase algal biomass and the deposition of fresh OM made increasingly available for bacterial respiration (Cole et al., 2000). As a result, the effect increased benthic respiration offsets $CO_2$ uptake

by primary producers, while extremely high influx of dissolved N can also favour microbial processes such as nitrification and denitrification which increase $CO_2$ evolution (Bogard et al., 2017).

Hydrological controls are known to be important regulators of water chemistry in small lentic systems (Reverey et al., 2016; Nitzsche et al., 2017; Peacock et al., 2019). Here we show that sites which received most of their inflow from snowmelt or groundwater, and which had short WRT supported supersaturated $CO_2$ concentrations (Fig. 2F). Such patterns may reflect

increased inputs of groundwater which are typically supersaturated with $CO_2$ (Macpherson, 2009). Long WRT is associated with larger, deeper systems. These sites are usually less influenced by the terrestrial-aquatic interface, take longer to concentrate the effect of any catchment-derived solutes (Junger et al., 2019), and have higher biotic assimilation of nutrients (Devito and Dillon, 1993; Fairchild and Velinsky, 2006). Larger waterbodies may also be able to better mediate stream or groundwater C inputs through longer chemical processing times and transformations. For example, agricultural reservoirs

with the highest WRTs tended to be hydrologically closed systems (E/I > 1) and any watershed derived DIC delivered from previous water sources is likely to be consumed by primary production which encourages atmospheric $CO_2$ uptake (Macrae



et al., 2004) Additionally, smaller waterbodies with shorter WRT can support higher rates of internal $CO_2$ production due higher rates of allochthonous DOC mineralisation (Weyhenmeyer et al., 2015; Vachon et al., 2017).

Groundwater delivery of DIC-rich porewater is the most likely hydrological source resulting in $CO_2$ enrichment of small

farm reservoirs. This mechanism is also suggested by the observation that higher reservoir $CO_2$ concentrations are predicted in high CEC soils Alkaline high CEC soils retain more calcium ions within clay particles which releases carbonates and bicarbonates into soil porewater (Kelley and Brown, 1934). Although regional snowmelt and groundwater have similar isotopic signatures (Pham et al., 2009; Jasechko et al., 2017), the positive correlation of $CO_2$ with alkalinity suggests groundwater as the main source. Edaphic sources of inorganic carbon can result in farm waterbodies accumulating dissolved

$CO_2$, bicarbonates, and carbonates, and therefore alkalinity, from the surrounding soils via groundwater discharge (Miller et al., 1985). Other studies have found strong evidence for groundwater inputs driving $CO_2$ supersaturation in small lentic systems (Perkins et al., 2015; Peacock et al., 2019) and watershed-derived alkalinity driving $CO_2$ supersaturation in lakes (Marcé et al., 2015).

Finally, landscape elevation had a significant external effect on reservoir $CO_2$ and may represent diverse weak controls

related to landscape setting. Lower $CO_2$ concentrations at higher elevations are common in 'perched' ecosystems with smaller contributing catchment areas (Diem et al., 2012) and low rates of allochthonous carbon influx (Rose et al., 2015). Conversely, waterbodies low in the landscape may receive more watershed C via groundwater influx due to topographical gradient (Winter and LaBaugh, 2003; van der Kamp and Hayashi, 2009). The effect of elevation could also be related to changes in vegetation composition within the local landscape, with the lowest lying catchments exhibiting higher abundance

of marginal wetland vegetation (Zhang et al., 2010) which favours higher inputs of terrestrial C (Magnuson et al., 2006; Abril et al., 2014).

### 4.2 Environmental drivers of $CH_4$ concentrations

The GAM suggested that $CH_4$ concentrations were primarily related to by internal biogeochemical processes and the influence of the hydrological regime. For example, factors related to water column productivity (DO, sediment C/N, DIN,

conductivity) had the most significant effect ($p$ <0.01), while some of the broader landscape features such as soil $K_{sat}$ and elevation had no significant effect on $CH_4$ levels. The nutrient status of waterbodies is often a primary driver of high $CH_4$ emissions in lakes, impoundments, and ponds (DelSontro et al., 2018; Peacock et al., 2019). Consequently, high nutrient availability is likely fuelling elevated values in both $O_2$ saturation and $CH_4$ (Fig. 3A). High $CH_4$ concentrations at low $O_2$ saturation reflects the development of anoxic habitats which favours methanogenesis (Huttunen et al., 2003;Bastviken et al.,

2004). This is likely the result of rapid biomass production which both enriches epilimnion with $O_2$ and depletes $O_2$ in the hypolimnion by providing fresh labile organic matter for decomposition.

In support of eutrophication-driven $CH_4$ production, our model indicated that high proportions of autochthonous organic matter in sediments were associated with elevated concentrations of $CH_4$ (Fig. 3B). Overall, sedimentary C/N ratios were in the range (8.5 to 13.4) expected for both phytoplankton and submerged macrophytes (Liu et al., 2018). This suggests that *in*





*situ* rather than terrestrial organic matter (OM), was the main source of C fuelling methanogenesis in these reservoirs. Strong associations of labile autochthonous C and $CH_4$ production in sediments (Due et al., 2010; Crowe et al., 2011) also suggests a direct link between eutrophication and $CH_4$ production in small farm waterbodies.

Unexpectedly, thermal stratification of the water column did not significantly influence $CH_4$ concentrations in small farm reservoirs (Fig. 3E). This finding contrasts with observations from other small waterbodies where limited mixing favours

$CH_4$ accumulation (Kankaala et al., 2013). Although some small systems exhibit diurnal mixing patterns with turnover at night (Glaz et al., 2016), the wide range of buoyancy frequency values (0.00 to 0.16) suggests that at least some farm reservoirs are continuously stratified, particularly in deeper ponds (Kankaala et al., 2013), as noted for $CO_2$ distributions (see above and Fig. 2D). Taken together, our findings suggest that variability in the biological production of $CH_4$ likely exerts a stronger influence over $CH_4$ concentrations across farm reservoirs than does physical mixing, and further supports the

hypothesis that the prevailing sediment and water chemistry are the primary controls of $CH_4$ concentrations.

Although the hydrological regime of small water bodies is rarely measured, we find that water source (rain, snow/groundwater) and reservoir retention time interact to influence $CH_4$ concentrations (Fig. 3F). In particular, $CH_4$ concentrations were lowest when WRT was long (>1 year) and water was derived mainly from snow or groundwater sources ($\delta^{18}O$ depleted). This may be due to a combination of reasons, including the prevalence of sulfate delivered from

groundwater (Pennock et al., 2010), dilution of waterbody from snow melt inflow, and sediments depleted in labile carbon due to longer biogeochemical processing times in the dams. The effect potential effect of sulfate limiting methanogenesis is in agreement with the strong negative relationship found between $CH_4$ and conductivity in our model (Fig. 3D). Sulfate makes up a large portion of the ionic composition of groundwater in the Prairie Pothole Region due to pyrite oxidation (Goldhaber et al., 2014). Clearly, the biological influence on $CH_4$ concentrations is less pronounced in these larger, low-flow

dams.

In contrast to the external drivers found for $CO_2$, local land use had a significant effect on $CH_4$ concentrations in farm reservoirs (Fig. 3I), with significantly higher $CH_4$ levels in cropland waterbodies than those in pasture. This finding contrasts with those from Australian farm reservoirs where diffusive $CH_4$ fluxes were 250% higher in reservoirs with livestock compared to crops, although the mechanisms responsible for observed differences were inconclusive (Ollivier et al., 2019).

Catchment land use regulates the physioco-chemical properties of ponds (Novikmec et al., 2016) by influencing the degree of local vegetative cover and associated influx of allochthonous C to waterbodies (Whitfield et al., 2011). Similarly, regions with crops undergo more intensive agricultural modification, with fertilisation, crop rotations, and mechanical disturbance of soil which all lead to greater nutrient runoff and soil erosion. In this case it's likely that $CH_4$ levels are more influenced by nutrient loading from the landscape which stimulates eutrophication (Huttunen et al., 2003), as suggested by the biotic

variables in our model (Fig. 3).





### 4.3 Emissions from farm reservoirs compared to other small waterbodies

To date, small waterbodies on farms have been shown to be large emitters of both $CO_2$ and $CH_4$ (Fig. 4). However, in our study we show that this is not always the case. Diffusive fluxes varied -21 to 466 and 0.14 to 92 mmol m$^{-2}$ d$^{-1}$ for $CO_2$ and $CH_4$, respectively. These findings are consistent with other small artificial waterbodies which are strong $CH_4$ sources that

exhibit a large range of variability from 0.02-33 mmol m$^{-2}$ d$^{-1}$ (Grinham et al., 2018a; Ollivier et al., 2019). The negative fluxes observed in our farm dams represents one of the few studied small waterbodies that exhibit $CO_2$ sink behaviour, with most showing net heterotrophy (Fig. 4). Although other studies have noted $CO_2$ sink behaviour in artificial ponds and reservoirs (Peacock et al., 2019; Ollivier et al., 2019), this is the first study to capture such a high proportion (>52%) of $CO_2$ uptake in such systems.

When $CO_2$ and $CH_4$ fluxes from small artificial waterbodies are compared with natural small waterbodies, no apparent trend exists in which group produces more or less carbon emissions (Fig. 4). Natural ponds and constructed waterbodies have a similar range in variability of mean fluxes for both gases, while wetlands exhibit some of the greatest within-study variability. Constructed waterbodies often have lower net $CO_2$ efflux, suggesting that these systems more often switch between net autotrophy and heterotrophy than small natural systems. Small artificial waterbodies have disproportionately

higher $CO_2$ and $CH_4$ emissions than other natural waterbodies due to the direct impact of agricultural and urban land use (Wang et al., 2017). However, analysis of the limited literature shows that is not the case. We suggest that the lack of a clear distinction between constructed and naturally-occurring small water bodies arises because of geographical variation in the relative importance of the diverse factors regulating carbon metabolism (Fig.s 2, 3).

When assessing the GHG impact of constructed waterbodies, it is important to consider the relative contribution to $CO_2$-

equivalent ($CO_2$-e) fluxes between $CO_2$ and $CH_4$. Here, $CH_4$ fluxes were converted to $CO_2$-e fluxes using the sustained-flux global warming potential over 100 years (Neubauer and Megonigal, 2015). Small natural ponds and wetlands have some of the highest $CO_2$-e emission rates, with particular importance of contributions from $CH_4$ (Fig. 5). On average our farm reservoirs had one of the highest $CH_4$ contribution to $CO_2$-e fluxes (74%), in agreement with the one other farm reservoir study (83%) of $CH_4$ contribution (Ollivier et al., 2019). This large contribution from $CH_4$ is similar to patterns recorded from

lakes and impoundments globally, where large freshwater bodies contribute to 75% of all $CO_2$-e efflux (DelSontro et al., 2018). Fortunately, because the factors that regulate $CH_4$ emissions are becoming better identified (Fig. 3), there exists the possibility that artificial wetlands can be constructed to minimize $CH_4$-related $CO_2$-e emissions and mitigate the overall large rate of $CO_2$-e emissions from agriculture (Robertson et al., 2000).

### 4.4 Minimising emissions: potential management solutions

A combination of factors, including landscape position, construction, and management, could optimize features to minimize carbon emissions from reservoirs and potentially enhance the carbon storage on farms. From our models, we suggest that key variables including the degree of water column stratification (buoyancy frequency), WRT, water source, land use, and





elevation are all suitable parameters for management. For example, strategizing landscape positioning to favour groundwater influx of sulfate to reduce methanogenesis. Creating deeper reservoirs will promote primary production through increased

water clarity (Dirnberger and Weinberger, 2005), facilitate $CH_4$ oxidation through the water column (Bastviken et al., 2008), and reduce the impact of watershed-derived solutes, terrestrial OM and benthic respiration. Additionally, deeper and larger artificial waterbodies tend to have lower nutrient concentrations due to longer processing times (Chiandet and Xenopoulos, 2016). Finally, modest increases in pH may further enhance $CO_2$ capture (Supporting Information), while having limited effect on $CH_4$ fluxes (Fig. 3).

Agricultural and urban waterbodies are highly susceptible to nutrient enrichment due to their direct proximity to intensified land uses. Reducing nutrient loading from the landscape will likely have one of the greatest impacts in minimising C emissions from farm dams given that both $CO_2$ and $CH_4$ were strongly predicted by inorganic N-species. In Australian farm reservoirs, for example, a 25% reduction of nitrates can reduce $CO_2$-e emissions by 50% (Ollivier et al., 2019). Similarly, removing direct livestock access to farm waterbodies will improve water quality overall through reducing direct DIN inputs

and dam infilling.

Nitrogen loading can also have a direct influence on nitrous oxide ($N_2O$), the third most potent greenhouse gas that can contribute substantially to $CO_2$-e emissions in farm systems (Robertson et al., 2000). The flux of $N_2O$ was constrained in our earlier study (Webb et al., 2019), which found a small $CO_2$-e sink (-89 to -3 mg $CO_2$ $m^{-2}$ $d^{-1}$) for the majority of these farm reservoirs despite high N concentrations. Similar to our $CO_2$ model, stratification and primary production were important

regulators in driving $N_2O$ uptake (Webb et al., 2019). Therefore, the potential to achieve net GHG sinks weighs mostly on the ability to reduce $CH_4$ emissions in these systems.

It is important to note that the $CH_4$ contribution to $CO_2$-e emissions is likely underestimated here as ebullition emissions were not measured. In farm reservoirs, ebullition flux can contribute >90% of total $CH_4$ emissions and is often highest in the smallest size classes (Grinham et al., 2018a). This reinforces that design and management strategies that focus on reducing

all pathways of $CH_4$ emissions will be most effective in curbing total $CO_2$-e emissions. Deeper farm dams with steep side slopes will likely be effective in reducing ebullition events due to a limited macrophytes, reduced bottom water temperature in summer, and supressed bubble release with higher water pressure (Joyce and Jewell, 2003; Natchimuthu et al., 2014; Grinham et al., 2018b).

**5 Conclusion**

Until recently, carbon emissions from small farm reservoirs have been an overlooked, yet potentially important source of $CO_2$ and $CH_4$ emissions within agricultural carbon budgets. To date, development of management strategies to reduce GHG emissions from waterbodies has been limited by lack of knowledge about the mechanisms regulating $CO_2$ and $CH_4$ production in these systems. By utilising adaptive modelling techniques across a broad range of environmental variables (abiotic, biotic, hydromorphological, landscape properties), we were able to explain a high degree of deviance in reservoir





$CO_2$ and $CH_4$ concentrations. We found that *in situ* water chemistry and local hydrological regime had the strongest impact on $CO_2$ and $CH_4$ concentrations. In agreement with previous studies, $CH_4$ fluxes were the largest contributor to $CO_2$-e emissions. However, in 19 reservoirs the net $CO_2$-e emissions were found to be sinks. We suggest that with optimal reservoir design and management the climatic impact of farm reservoir C-emissions has the potential to be a carbon net sink. To further develop farm reservoir management practices that are locally effective, we express a need for more widespread farm

waterbody GHG measurements across the globe to cover other continents and land uses.

**Data availability:** All data used in the models is available online in a GitHub repository (https://github.com/JackieRWebb/Dugouts-CO2-CH4). Public access to this repository will be made available upon publication and a DOI will be generated at this time.


**Supplement:** The supporting information related to this study will be published online.

**Author contributions***:* J.R.W., G.L.S., P.R.L., H.M.B., and K.F. designed research; J.R.W. performed research and wrote the paper; H.M.B. contributed new reagents/analytic tools; H.A.H., P.R.L., G.L.S., and K.F. contributed towards ideas and

data analysis; K.R.H performed GIS analysis; and G.L.S. developed models.

**Competing interests:** The authors declare no competing interests

**Acknowledgements:** Financial support for data collection and analyses were provided in part by Government of

Saskatchewan (Award 200160015), Natural Sciences and Engineering Research Council of Canada Discovery grants (to K.F., G.L.S., H.M.B., and P.R.L.), the Canada Foundation for Innovation, University of Regina. We thank Jessica Bos, Corey McCowan, Lauren Thies, Ryan Rimas, and Nathanael Bergbusch for fieldwork assistance and all landowners for their generous cooperation in volunteering their reservoirs for this research.

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





**Tables and Figures**

**Table 1: Farm reservoir and landscape physical, hydrological, and chemical characteristics of the study sites (n = 102)**

| | Units | N | Mean | Median | Min | Max |
|---|---|---|---|---|---|---|
| Area | $m^2$ | 101 | 1,312 | 1,040 | 158 | 13,900 |
| Depth | m | 102 | 2.08 | 2.10 | 0.18 | 5.10 |
| Buoyancy frequency | $s^{-2}$ | 99 | 0.01 | 0.005 | 0.00 | 0.03 |
| $\delta^{18}O$ inflow | ‰ | 101 | -13.37 | -13.33 | -19.39 | -8.40 |
| Evaporation to inflow | | 101 | 0.46 | 0.43 | 0.04 | 1.58 |
| Water residence time | Years | 100 | 0.76 | 0.66 | 0.08 | 2.51 |
| $CO_2$ | μM | 101 | 42.2 | 14.6 | 1.3 | 326.1 |
| $CH_4$ | μM | 101 | 4.3 | 1.9 | 0.1 | 54.5 |
| Temperature | °C | 102 | 20.1 | 19.9 | 15.7 | 29.5 |
| Dissolved $O_2$ | % | 102 | 92.6 | 88.9 | 2.3 | 344.0 |
| Salinity | ppt | 102 | 0.9 | 0.5 | 0.1 | 8.6 |
| pH | | 102 | 8.75 | 8.75 | 6.95 | 10.19 |
| Chlorophyll a | μg $L^{-1}$ | 102 | 99.1 | 36.9 | 2.2 | 2,483 |
| $NH_3$ | μg N $L^{-1}$ | 100 | 354.7 | 100.0 | 10.0 | 5,930 |
| $NO_x$ | μg N $L^{-1}$ | 98 | 196.6 | 34.1 | 1.2 | 3,188 |
| TP | μg P $L^{-1}$ | 98 | 285.2 | 80.0 | 8.7 | 6,480 |
| TN | μg N $L^{-1}$ | 98 | 3,082 | 2,360 | 417.5 | 14,280 |
| DOC | mg C $L^{-1}$ | 99 | 31.8 | 29.3 | 4.6 | 90.4 |
| Sediment organic carbon | % | 101 | 5.2 | 3.9 | 0.6 | 31.4 |
| Sediment organic nitrogen | % | 101 | 0.6 | 0.4 | 0.1 | 2.8 |
| Alkalinity | mg $L^{-1}$ | 96 | 245.4 | 219.2 | 71.0 | 755.5 |
| Soil CEC | M-eq $100g^{-1}$ | 98 | 24 | 24 | 10 | 180 |
| $K_{sat}$ | cm $hr^{-1}$ | 102 | 9.9 | 5.0 | 0.0 | 39.7 |
| Elevation | m | 102 | 627.6 | 598.0 | 484.0 | 997.0 |





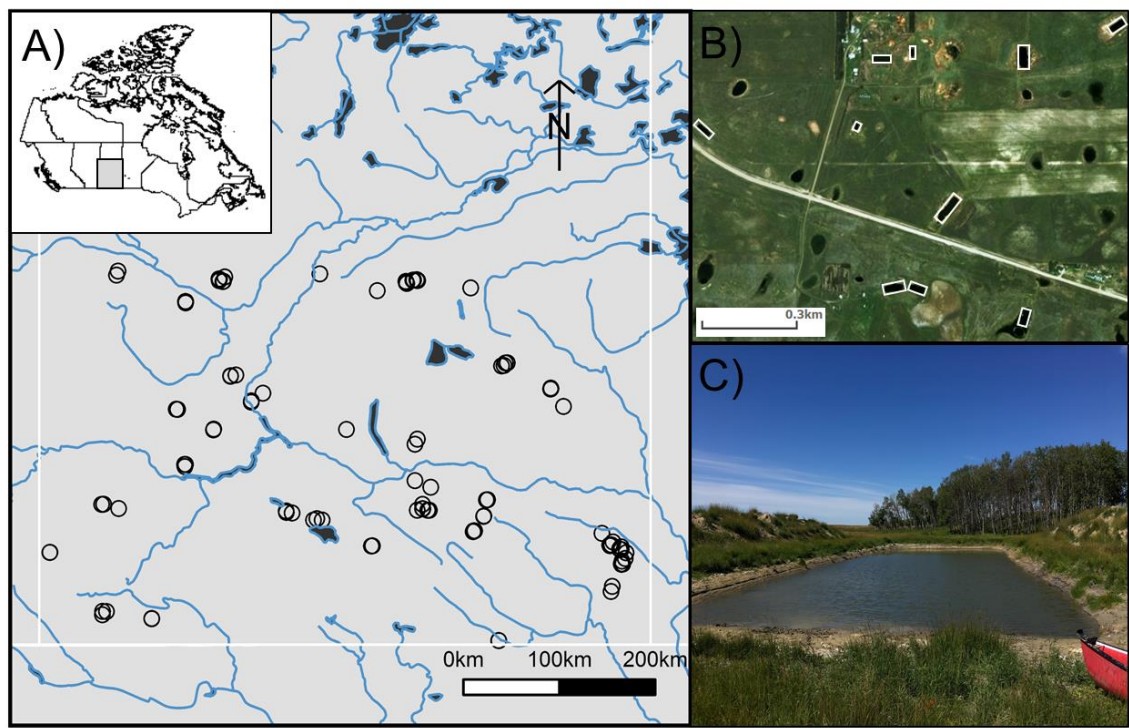


**Figure 1: A) Map of southern Saskatchewan in Canada showing the distribution of studied farm reservoirs, B) aerial image showing 10 farm reservoirs delineated by white rectangles within a 1 km² area, and C) general size and shape of farm reservoirs with two characteristic side mounds of excavated materials.**







**Figure 2: Response patterns farm reservoir CO₂ concentrations with abiotic, biotic, hydromorphological, and landscape variables based on GAMs. CO₂ was best estimated by a combination of a) DO saturation, b) alkalinity, c) NOx, d) buoyancy frequency, e) interaction between δ_I and WRT, f) soil CEC, g) and elevation, with soil salinity (h) not significant. Model deviance explained was 65.7%. The response patterns shown are the partial effect splines from the GAM and shaded area indicated 95% credible intervals. See Table S4 for summary of model statistics.**





**Figure 3: Response patterns farm reservoir CH₄ concentrations with abiotic, biotic, hydromorphological, and landscape variables based on generalised additive models (GAMs). CH₄ was explained by a combination of a) DO saturation, b) sediment C/N, c) DIN, d) conductivity, e) buoyancy frequency (not significant, f) interaction between δ_I and WRT, g) soil Ksat (not significant), h) elevation (not significant), and i) local land use. Model deviance explained was 74.1%. The response patterns shown are the partial effect splines from the GAM and shaded area indicated 95% credible intervals. See Table S5 for summary of model statistics.**


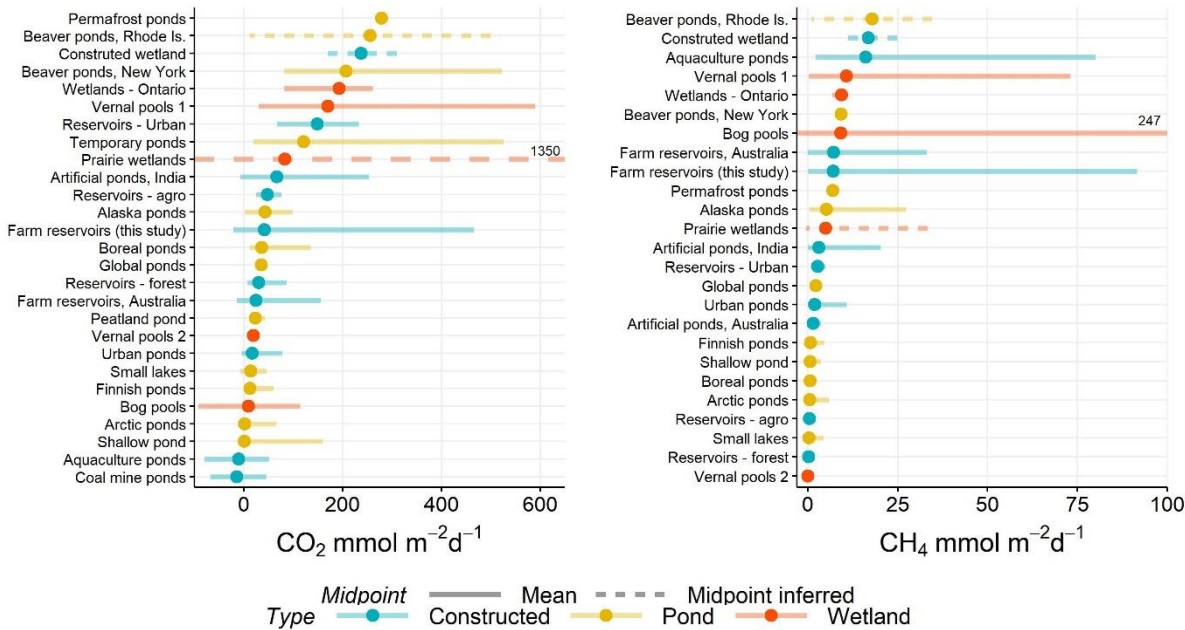

**Figure 4: Range of CO$_2$ and CH$_4$ (diffusive) fluxes observed in natural and constructed small (<0.01 km$^2$) waterbodies, including this study (farm reservoirs). Dots represent the mean reported in each study and error bars the range. If no mean value was reported, then the midpoint was inferred as the middle of range (dashed lines). All data is from the published literature and references can be found in the Table S6.**



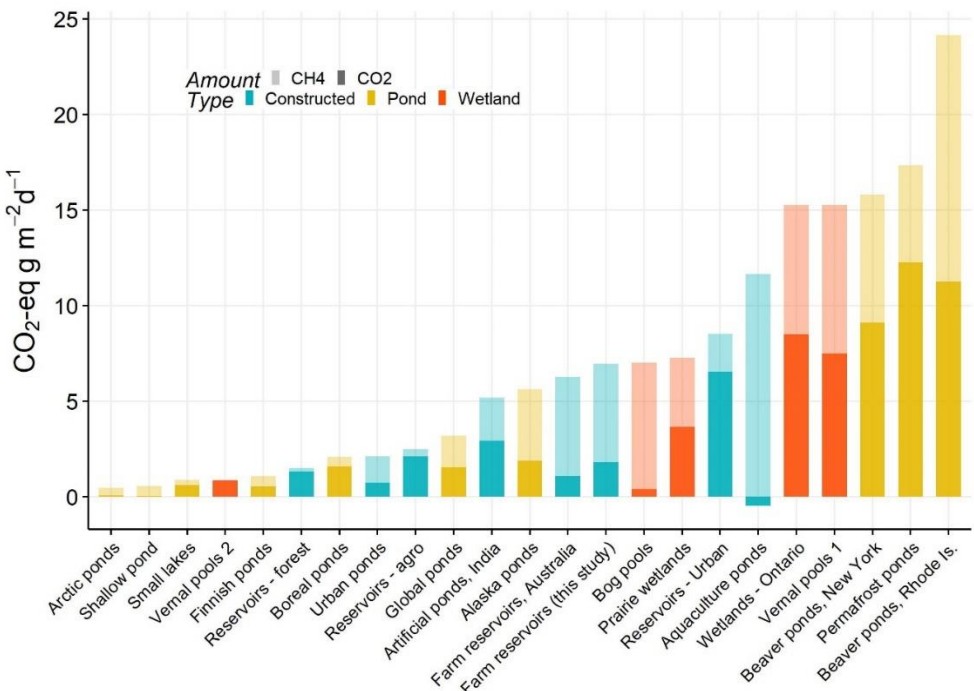

**Figure 5: Total CO₂ equivalent fluxes of CO₂ and CH₄ (diffusive) measured in natural and artificial small waterbodies (<0.01 km². CO₂-e fluxes were calculated based on 100 year sustained-flux global warming potentials in Neubauer and Megonigal (2015). Relative proportions of each gas are indicated by shading, and waterbody type is given by colour. All data is from the published literature and references can be found in the Table S6.**