# Peer review of "Regulation of carbon dioxide and methane in small agricultural reservoirs: Optimizing potential for greenhouse gas uptake"

_Biogeosciences, 2019_

## Referee Comment (RC1) · Anonymous Referee #1 · 12 Aug 2019

The paper by Webb et al presents $CH_4$ and $CO_2$ data from 101 farm ponds. Alongside these GHG measurements are an impressive array of variables of water chemistry, hydrological characteristics, and landscape attributes. The authors investigate these variables as drivers of the GHG emissions. The paper is well written and I enjoyed reading it. It is within the scope of BG, and presents novel data insomuch as the fact that more pond GHG data is needed (and this point was explicitly raised in the recent IPCC refinement). If small, artificial waterbodies can be designed to minimise $CH_4$ emissions, and to act as $CO_2$ sinks, then this could lead to them acting as natural climate solutions.

Methods and analysis are well explained with sufficient detail, and the results support the conclusions. Presentation is good, language is fluent, abstract is suitable. The work is mostly well referenced (I suggest two older references of farm pond emissions that the authors may have missed). I particularly enjoyed reading the succinct and to-the-point results section, which was enough to get the authors' points over without endlessly writing numbers out, as so many results sections do.

The one thing I find lacking from the paper is a visual presentation of the underlying $CO_2$ and $CH_4$ data, and in my comments I suggest a way to address this. I think it is important that readers are offered an easy way to understand the variation in the GHG data across all 101 waterbodies.

I suggest the paper is acceptable following minor revisions. Below are my detailed comments.

L29. "Small waterbodies have recently been recognised as substantial contributors to global carbon emissions from inland waters." This is true, and missing from somewhere in the introduction (and discussion) is a mention that the recent 2019 IPCC Refinement explicitly addresses the issue of $CH_4$ emissions from artificial ponds. The Refinement can be found at the link below, and the relevant chapter is in vol. 4 (AFOLU), chapter 7 (Wetlands). The emission factor given for artificial ponds is 183 kg $CH_4$/ha/yr, but there is currently not enough data to disaggregate pond emissions by climate zone. How does your data compare to this emission factor? https://www.ipcc-nggip.iges.or.jp/public/2019rf/index.html

L36. It's worth noting the recent paper by van Bergen et al who measured $CH_4$ (including ebullition) and $CO_2$ emissions, and C burial of an urban pond. Ideally we need studies that quantify GHG emissions and C burial, so the net balance can be calculated. van Bergen, T.J., Barros, N., Mendonça, R., Aben, R.C., Althuizen, I.H., Huszar, V., Lamers, L.P., Lürling, M., Roland, F. and Kosten, S., 2019. Seasonal and diel variation in greenhouse gas emissions from an urban pond and its major drivers. Limnology
and Oceanography.

L60. "Currently, only three studies have comprehensively assessed C fluxes from small agricultural reservoirs." What does "comprehensively" mean in this case? These three studies are slightly different – Ollivier et al did not measure ebullition whilst the other two studies did. Ollivier et al and Paneer Selvam et al were 'snapshot' studies whilst Grinham included some temporally repeated measurements (but didn't measure CO2). So are they all comprehensive really? I accept this is a minor point of language but it does matter. Additionally, there are two other papers that have measured farm ponds. Stadmark et al made repeated measurements of CH4 and CO2 emissions from agricultural ponds created to retain N: Stadmark, J. and Leonardson, L., 2005. Emissions of greenhouse gases from ponds constructed for nitrogen removal. Ecological Engineering, 25(5), pp.542-551. There is also data in an old and rather blandly titled paper from two farm ponds. Baker-Blocker, A., Donahue, T.M. and Mancy, K.H., 1977. Methane flux from wetlands areas. Tellus, 29(3), pp.245-250.

L62. "Large fractions of CH4 being released." Fractions seems like an odd and unsuitable word. Change for "volumes", "amounts", "quantities", etc?

L80. The study region occupies a large area, but seeing as temperatures are given it would also be good to give a value (or range) for annual precipitation. Reading on, I see the results says "precipitation ∼60% less than the long-term climate average of 390 mm in Regina." Please give the value in the methods.

L86. It says 101 ponds were sampled, but in table 1 some variables have N = 102. Where does 102 come from?

L113, L118. Floating chambers are not "incubations". This word should be altered to something like "deployments" or similar.

L121. It says DO was measured in mg/l but in table 1 it is given as %. The methods text should be amended to % instead.

L149. Inflow is mentioned here. Do these systems have inflows? Is water pumped in for storage, or do they simply collect rainwater?

L183. "To avoid multicollinearity, correlation coefficients between pairs from Pearson linear correlation tests was used to guide covariate choice before model fitting." This is vague. Did you use a Pearson correlation coefficient of a certain value to decide when multicollinerity was present?

L197. Something I desperately miss from the paper is a figure allowing the reader to visualise the raw CH4 and CO2 data and its distribution. I strongly advise the addition of a figure to show this. It could take numerous forms, such as a scatter plot of CH4 vs CO2 for all 101 ponds, or a box plot of GHGs (grouped by pond size, or pasture vs cropland), or even a bar plot showing individual concs for 101 ponds (large and unwieldy perhaps, but visually useful). Reading on I see figure.3 has a very small land-use graph, but I think a more obvious, up-front figure would be better.

Fig 2 and fig 3. In part this relates to my point above. Wouldn't these figures be improved by adding the underlying data points on to these figures as a scatter? That way the reader can see the model, and the raw data. It would help the reader visually determine the robustness of the models.

L210. "CO2concentrations displayed a positive response with…NOx" Whilst the upper 95% credible interval continues to increase, the black line presumably suggests that CO2 decreases at the highest NOx levels. Is there a mechanism that can explain this?

Figure 3 has a land use graph, but figure two doesn't. Even if there is no difference in CO2 between land use a figure would still be interesting to see, and there is room for an extra panel at the bottom right anyway.

For the land use panel in figure 3, the categories are pasture, livestock and cropland. However, line 87 in the methods only mentions pasture (n = 80) and cropland (n = 21). Where do these livestock ponds come from?

L224. "Our comprehensive spatial analysis revealed wide variations among CO2 and CH4 concentrations between farm reservoirs" As per my previous comment, there's currently no easy way to assess this until the raw data is more visible in a figure.

L227. "CH4 was most correlated by internal abiotic and biotic mechanisms" Should this not be "most correlated with"?

L282. "Additionally, smaller waterbodies with shorter WRT can support higher rates of internal CO2 production due higher rates of allochthonous DOC mineralisation" Needs amending to read "due to"

L285. "This mechanism is also suggested by the observation that higher reservoir CO2concentrations are predicted in high CEC soils Alkaline high CEC soils retain more calcium ions within clay particles which releases carbonates and bicarbonates into soil porewater" It seems like something has gone awry in the writing here, and this should be two sentences or some words need removing.

L331. "The effect potential effect of sulfate" The first "effect" needs deleting

L336. "In contrast to the external drivers found for CO2, local land use had a significant effect on CH4 concentrations in farm reservoirs (Fig.3I), with significantly higher CH4 levels in cropland waterbodies than those in pasture. This finding contrasts with those from Australian farm reservoirs where diffusive CH4 fluxes were 250% higher in reservoirs with livestock compared to crops," I find this section of the discussion interesting. As the authors write, the intensive agricultural practices associated with cropland could be expected to result in elevated CH4 concentrations. Conversely, pasture/livestock emissions would depend on the system (intensive or extensive), livestock, etc. Intensive grassland systems could easily result in high emissions, whilst low-level grazing might result in emissions being less than those from cropland. So cropland > grassland and grassland < cropland are both explicable it seems to me.

Figure 4 and fig. 5. The study by Grinham et al of Australian ponds is referenced in

the text but doesn't seem to be included in these figures. Is there any reason their data was left out?

L365. "Here, CH4 fluxes were converted to CO2-efluxes using the sustained-flux global warming potential over 100 years" I am not familiar with this metric, and suggest a few lines are included in the methods as to what it is and how it is calculated.

Section 4.4. What (if any) vegetation colonises these pools? Is there no role for encouraging certain plant species that might promote C uptake? For instance, Moore & Hunt say: "The carbon sequestration assessment of constructed stormwater wetlands and ponds suggests that emergent vegetation is a significant source to the soil carbon pool (compared to allochthonous sources) and a critical component of carbon sequestration in these systems." Moore, T.L. and Hunt, W.F., 2012. Ecosystem service provision by stormwater wetlands and ponds–A means for evaluation?. Water research, 46(20), pp.6811-6823.

L392. "The flux of N2O was constrained in our earlier study (Webb et al., 2019), which found a small CO2-e sink (-89 to -3 mg CO2m-2d-1) for the majority of these farm reservoirs despite high N concentrations." Something of a diversion here, but doesn't this depend on how the data are interpreted though? In your earlier study the median N2O flux was negative, but the mean was positive (with 33% of ponds emitting N2O), whilst in this study (figs 4 and 5) you present mean CH4 and CO2. There's probably a debate to be had concerning what average is most appropriate to use, but note the IPCC Refinement used a mean value calculated from log-transformed values.

---

## Referee Comment (RC2) · Anonymous Referee #2 · 19 Aug 2019

This paper describes $CO_2$ and $CH_4$ concentration measurements made during the summer season on 101 farm reservoirs in an agricultural region of Saskatchewan, Canada. The authors then use a series of floating chamber measurements to infer diffusive fluxes of these two greenhouse gases at the pond surface via estimations of gas transfer. The authors also collect data on a number of abiotic and biotic landscape/waterbody characteristics that may help predict farm pond GHG concentrations. They then use general additive modeling to describe controls on waterbody concentration. While not currently emphasized, this paper follows up on a previous article that described novel $N_2O$ uptake dynamics in these same ponds. The authors emphasize a few findings: 1) more than half of farm ponds are net $CO_2$ sinks, 2) some (19%)

farm ponds are net CO2-eq sinks when looking at diffusive emissions, 3) CO2 concentrations are governed most by hydrology/landscape position, 4) CH4 emissions are governed most by autochthonous production.

The current framing of this paper is difficult for me to digest given the complete lack of any CH4 ebullition measurements from these systems (and given that fluxes were estimated based on highly uncertain estimates of gas transfer). While the authors acknowledge that their estimates of CO2-eq emissions are likely low due to the lack of ebullition measurements, this is done at the very end of their paper. I think this point should be made sooner as it is an important detail that influences the interpretation of their findings. The relative contribution of ebullition to total methane flux can vary widely from system to system and the controls on the proportion of methane flux that is ebullitive are not well understood (Deemer et al. 2016 BioScience). It would be helpful to know if the authors observed any evidence of ebullition events during their floating chamber surveys? How much ebullition would have to be observed to push the net CO2-eq sink systems towards net-source? Also, what is the uncertainty in sink vs. source estimations due to uncertainty in system gas transfer velocity? To this same end, it is difficult to see the 19% of systems that are net CO2-eq sinks by looking at the authors' figures. Is this because the net CO2-eq sink is very small? For example, Figure 4 does not seem to show that over 50% of the systems in your study were net CO2 sinks. I suggest adding a zero line to your figures and possibly creating an additional figure that shows fluxes site-by-site for the farm ponds in your study. The visual aids currently offered for showing the distribution of your own dataset are sort of overshadowed by a comparison with the broader literature.

Also, while I am not very familiar with GAMs, I found this analysis a bit opaque and difficult to interpret as currently described. For example, were both N and P variables put into the model and NOx/DIN came out as more important? Also, how were the variables plotted in figures 2 and 3 selected? From what I can gather, you have plotted more than just the variables in the best model. For the sake of discussion, it would be

nice to see a consistent set of variables and their relationship to both CH4 and CO2.

To me, the more novel part of this data set is the high fraction of ponds that are net CO2 sinks. This is also a finding that is most strongly backed by the data that was collected since the conclusion doesn't rely as much on gas transfer estimates and since CO2 ebullition is typically an extremely small fraction of total CO2 emission. The extent of the CO2 sink in these small agricultural ponds could be compared to the lesser extent reported in the global data set of artificial reservoir GHG dynamics (Deemer et al. 2016). It is also interesting that the CO2 sink seems to scale more with landscape and hydrological factors than with ecosystem productivity. While multiple other studies have already emphasized the potential importance of nutrient management/eutrophication on lake, pond, and reservoir methane emissions (see Beaulieu et al. 2019 for a very recent global scale discussion), the findings you present in this paper suggest that landscape placement of farm reservoirs may help buffer GHG emissions independent of trophic status (via carbonate buffering and groundwater DIC chemistry dynamics). See paper by Pacheco et al 2013 in Inland Waters (which asks if eutrophication can reverse the aquatic C budget). To this end, it would also be nice to see plots comparing emission by land use for both CH4 and CO2 (right now the plot is only shown for CH4).

The comparison between human-made and natural waterbodies is also interesting and novel. I think it would be good to more thoroughly introduce this question/concept (that the systems might fundamentally differ from each other) earlier in the paper and then come back to it in the discussion. A good reference for comparing human-made and natural waterbodies is Hayes et al. 2017 L&O Letters as well as Doubek & Carey 2017 Inland Waters.

Line by Line Edits

Line 18: add "surface" before "concentrations"

Lines 20-21: this is a little misleading since pH was actually a better predictor

Lines 23-24: state the timescale over which you are calculating CO2-equivalents

Line 26: bringing up depth doesn't seem appropriate here since depth didn't come out as a significant predictor variable in your models

Line 30-31: Holgerson and Raymond 2016 didn't look at ebullition

Line 45-46: Also check out Couto and Olden 2018... there aren't really global papers that distinguish surface area of small farm reservoirs/ponds from small hydropower.

Lines 46-47: I suggest listing out numbers of reservoirs by country since the current phrasing is difficult to interpret. Either that or use a word like "collectively" to indicate that 8 million is the sum across multiple countries.

Line 51: What does It mean to create reservoirs at a rate of up to 60% of standing stock? I'm a bit confused by this wording.

Lines 56-57: It is a bit awkward to suggest that eutrophication results in potent CO2 release since autochthonous production actually works to fix CO2 (see Pacheco et al. 2013).

Lines 76-77: I suggest clarifying: you are identifying drivers of surface water concentration, not total flux. Although these are related, they are not the same thing.

Lines 86-87: How did you select your sites? Randomly?

Lines 197-202: What were N:P ratios like in these systems?

Results section: I suggest including a summary of the fluxes you estimate (and associated gas transfer rates from the floating chamber surveys). Can you estimate how variability in k might affect variability in your flux estimates? Are there cases where you have both a floating chamber and a concentration based estimate of flux? How much did these differ from each other?

Line 227: change "by" to "with"

Line 246: Not a complete sentence.

Lines 261-262: This doesn't seem like a very satisfying explanation to me. Is it also possible that differing hydrology leads to the more stratified systems also being the ones that are higher in CO2?

Line 269: add "of" between "effect" and "increased"

Line 270: Nitrification doesn't produce CO2; it is an autotrophic process.

Line 272: This is a pretty vague topic sentence. It would be helpful to be a little more specific.

Line 303: get rid of "by"

Lines 306-307: Deemer et al. 2016 and Beaulieu et al. 2019 are also good references here.

Lines 312-315: Higher CH4 from higher C:N sediments suggests more (not less) important role for allochthonous C right?

Line 318-319: I would expect thermal stratification to influence bottom water CH4 concentration more than surface water CH4, but you only have surface water concentrations in your model.

Line 331: Get rid of second "effect"

Line 334-335: Avoid using the word "clearly". Also, it would be helpful to show the relationship between CH4 and salinity in your Figure 3 to support this discussion.

Lines 365-366: State the actual factor that you used here too. Was it 34?

Lines 392-393: It seems like it would be nice to mention this parallel study earlier in your paper and give it a bit more discussion.

Lines 378-383: This all seems very speculative. As do lines 400-403.

---

## Author Response (AR1)

**Associate Editor Decision: Reconsider after major revisions (16 Sep 2019) by Ji-Hyung Park**

Comments to the Author:

Thank you for providing detailed responses to the comments and suggestions offered by two reviewers.

Both reviewers recognized the scientific value and novelty of your manuscript, but the second reviewer also raised several critical issues. I agree that you need to pay more attention to uncertainties in estimating CO2 and CH4 fluxes when you evaluate the sink or source capacity of the studied reservoirs. I thought that the manuscript would require a substantial revision to address all the raised issues and a number of other comments, so I recommend 'reconsider after major revisions'.

**Response**: We thank the Associate Editor for these suggestions and consideration of this manuscript after revisions. We have addressed each comment below in further detail.

With regard to your assessment of net CO2 eq sink (19%), I would suggest that you consider uncertainties associated with CH4 loss via ebullition (as suggested by the second reviewer) and potential temporal (both diurnal and seasonal) variations in CO2 to provide ranges of estimates rather than one single estimate. You measured CO2 only in summer months, so you may have different (probably higher) values in other seasons due to changes in primary production. Please refer to other studies (or your own studies if you have) to estimate potential seasonal variations. You did not provide any detail about sampling frequency and time (Once per sampling, per season? Or repeated samplings to cover diurnal variation?). It would be quite misleading if you provide one single value out of uncertain estimates even though your sampling did not cover seasonal and diurnal variations.

**Response**: We followed your suggestion and assessed the range in $CO_2$-e sink capacity of our farm reservoirs based on preliminary seasonal data. Firstly, our diurnal data is greatly limited to only four sites, yet suggests far less variation than seasonal. The plots below show $CO_2$ and $CH_4$ concentrations at time = 0 (10 am) versus the average concentration over the 24 hr time series. Note that 4 sites were sampled over a 24-h period, and gases were collected every 6 hours (5 samples). We show that values collected at 10 am were not systematically higher or lower than the mean daily concentrations for a given site, suggesting low variability. This was likely due to the high alkalinity (a dominant characteristic for most farm reservoirs in this survey), buffering potentially large $CO_2$ fluctuations.

[Figure]

Therefore, the following paragraph has been added to the discussion.

*"On average, 8% of farm reservoirs were acting as $CO_2$-e sinks on the range of -0.6 to 79 g $CO_2$ $m^{-2}$ $d^{-1}$ during the time of sampling. This number offers a snapshot of the potential for farm reservoirs to act as a net $CO_2$-e sink and it is important to consider how seasonal variation influences the GHG sink/source status. Preliminary data on seasonal variation in $CO_2$ and $CH_4$ concentrations from a smaller number of farm reservoirs indicate variation (represented as the standard deviation related to the mean), ranging between 20 to 200% and 40 to 200% for $CO_2$ and $CH_4$, respectively. Here, this variation represents monthly sampling between the periods of ice melt and ice formation on lakes in Saskatchewan. Applying the average observed seasonal variation of 78% and 93% to our current spatial dataset suggests that $CO_2$-e emissions from farm reservoirs may vary between -1.7 and 150 g $CO_2$ $m^{-2}$ $d^{-1}$, or 0 to 44% as acting net $CO_2$-e sinks. Further study into the consistency of potential farm reservoir $CO_2$ sinks on the temporal scale is required to better assess the overall GHG impact."*
Line 400

Because we have not undertaken any direct ebullition measurements, we feel that providing an assessment of uncertainties associated with ebullition is too speculative to apply to this quantitative dataset. Instead we highlight the importance of measuring this pathway to further inform management strategies and design:

*"It is important to note that the $CH_4$ contribution to $CO_2$-e emissions is likely underestimated here as ebullition emissions were not measured. In farm reservoirs, ebullition flux can contribute >90% of total $CH_4$ emissions and is often highest in the smallest size classes (Grinham et al., 2018a). However, the sporadic nature of this pathway remains difficult to constrain for one single type of waterbody and may be a minor contributor in reservoirs and ponds > 3-5 m deep (Joyce and Jewell, 2003; DelSontro et al., 2016). This reinforces that design and management strategies that focus on reducing all pathways of $CH_4$ emissions will be most effective in curbing total $CO_2$-e emissions. Deeper farm dams with steep side slopes will likely be effective in reducing ebullition events due to a limited macrophytes, reduced bottom water temperature in summer, and supressed bubble release with higher water pressure (Joyce and Jewell, 2003; Natchimuthu et al., 2014; Grinham et al., 2018b)."*
Line 401

Finally, we have added additional information on the frequency and timing of sampling for this study:

*"Each site was sampled once during this period, between the daylight hours of 10:00 to 15:00."* Line 96

Considering the critical role of phytoplankton in reservoir CO2 budgets, you might also need to provide more descriptions and discussion on the relationship between Chl a and CO2. It appears that your model (and also your discussion) does not consider this important relationship. Please check and discuss any lack or hidden relationship between Chl a and CO2 to assess the role of phytoplankton as a CO2 sink, particularly in relation to nutrient levels in the studied reservoirs (for instance, in lines 198-199 you can provide more information about how CO2 varies with Chl a and nutrient levels). Your discussion on nutrient control over phytoplankton and CO2 levels (lines 262-270) focuses on the positive relationship between N and CO2. Please refer to other studies reporting various relationships between nutrients (both N and P) and phytoplankton uptake and release of CO2 (and CH4) to provide a more in-depth discussion of the observed patterns (your succinct data presentation does not allow readers to find out detailed information on this topic).

**Response**: We agree that autotrophic activity plays an active role in reservoir $CO_2$ budgets. The role of phytoplankton was initially tested using the parameter chlorophyll *a* (a measure of phytoplankton biomass) in the correlation tests for $CO_2$. Readers can find this presentation of the data in Supplementary Tables S1 and 2. It had a significant relationship with dissolved oxygen (DO), both representing the role of autotrophic activity. Because DO represented a more direct measure of

primary productivity at the time of sampling, and was more significantly correlated with reservoir $CO_2$ concentration, this parameter of primary production was selected for the final model, rather than Chl *a*.

- Line 22: It is not clear which optimal design and management can minimize GHG impact. Please elaborate on the implication of your findings in the context of GHG emission mitigation. You stated "evaluating the potential for reservoir design to minimize CO2-equivalent (CO2-e) emissions (line 71) as a primary goal of your study. However, as you mentioned in the following sentence ("By identifying the driving characteristics of farm dams that support reduced C emissions, our findings provide the first step to developing management strategies to help minimize farm carbon emissions."), your results appear to provide some baseline information that could be useful in opting for emission mitigation strategies. Because this baseline information is not specific enough to suggest "the potential for reservoir design to minimize" GHG emissions, a more cautious wording would help readers grab some practical ramifications of your scientific findings.

**Response**: The sentence in the abstract on design and management has now been elaborated to read:

*"From our models, we show that the GHG impact of farm reservoirs can be greatly minimised with overall improvements in water quality and consideration to position and hydrology within the land scape."* Line 25

We have also revised the wording in the study goal sentence to read:

*"Our aim was to identify the key environmental conditions regulating $CO_2$ and $CH_4$ fluxes, and to evaluate this baseline data in the context of emission mitigation strategies."* Line 76

- Line 73 (& 178-):. Is this (GAMs) a new approach proposed in this study? Please clarify whether you propose this approach here for the first time or simply follow other studies (then cite relevant references)

**Response**: GAMs are a fairly standard modelling tool in ecology statistics (Pedersen et al., 2019. https://doi.org/10.7717/peerj.6876). We have added the following statement when introducing GAMs in methods:

*"GAMs are not constrained by prescribed assumptions associated with parametric models such as linearity of link-scale effects in generalized linear models. Instead, the functional form of the partial relationships between covariates and the response are determined from the data. The more flexible modelling approach is useful where the effects of covariates on the response are non-linear and has been applied to complex aquatic datasets assessing GHGs (Wiik et al., 2018; Webb et al., 2019)."* Line 197

- Line 17 "address and manage their potential importance" – Please specify what specific importance you want to "address and manage" (?).

**Response**: "…in agricultural GHG budgets" has been added to the sentence (Line 17).

- Line 22 "eutrophication-driven CH4": Do you mean "eutrophication-driven production of CH4"?

**Response**: Statement has been revised to read "…a positive association between eutrophication and $CH_4$ production".

- Lines 28-54: These two paragraphs may be reversed in order.

**Response**: Corrected.

- Line 172 "NOx": Have you defined this earlier? Please note that NOx usually refers to nitrogen oxides in environmental science.

**Response**: We have added the following definition in parenthesis after first mention of $NO_x$ here: "…($NO_2 + NO_3$)", Line 191

- Figs. 4-5: Can you make the data of the label "Farm reservoirs (this study)" stand out by using some special symbol or color?

**Response**: We have now highlighted this study in bold on Figures 5 and 6:

[Figure]

[Figure]

End of Associate Editor response

**Anonymous Referee #1**

The paper by Webb et al presents CH4 and CO2 data from 101 farm ponds. Alongside these GHG measurements are an impressive array of variables of water chemistry, hydrological characteristics, and landscape attributes. The authors investigate these variables as drivers of the GHG emissions. The paper is well written and I enjoyed reading it. It is within the scope of BG, and presents novel data insomuch as the fact that more pond GHG data is needed (and this point was explicitly raised in the recent IPCC refinement). If small, artificial waterbodies can be designed to minimise CH4 emissions, and to act as CO2 sinks, then this could lead to them acting as natural climate solutions.

Methods and analysis are well explained with sufficient detail, and the results support the conclusions. Presentation is good, language is fluent, abstract is suitable. The work is mostly well referenced (I suggest two older references of farm pond emissions that the authors may have missed). I particularly enjoyed reading the succinct and to-the-point results section, which was enough to get the authors' points over without endlessly writing numbers out, as so many results sections do. The one thing I find lacking from the paper is a visual presentation of the underlying CO2 and CH4 data, and in my comments I suggest a way to address this. I think it is important that readers are offered an easy way to understand the variation in the GHG data across all 101 waterbodies. I suggest the paper is acceptable following minor revisions. Below are my detailed comments.

**Response**: We thank the reviewer for their positive review and their constructed comments and suggestions offered. Detailed responses to the comments are addressed in blue font below.

L29. "Small waterbodies have recently been recognised as substantial contributors to global carbon emissions from inland waters." This is true, and missing from somewhere in the introduction (and discussion) is a mention that the recent 2019 IPCC Refinement explicitly addresses the issue of CH4 emissions from artificial ponds. The Refinement can be found at the link below, and the relevant chapter is in vol. 4 (AFOLU), chapter 7 (Wetlands). The emission factor given for artificial ponds is 183 kg CH4/ha/yr, but there is currently not enough data to disaggregate pond emissions by climate zone. How does your data compare to this emission factor? https://www.ipccnggip.iges.or.jp/public/2019rf/index.html

**Response**: We appreciate the reviewer raising awareness of the latest IPCC estimate. The following sentence has now been added to the introduction:

"*The recent 2019 IPCC Refinement has assigned a CH$_4$ emission factor of 183 kg ha$^{-1}$ yr$^{-1}$ to constructed waterbodies, however data is greatly limited, both geographically and in number (n = 68), that climatic-zone emission factors cannot be estimated (IPCC, 2019).*" Line 61

We also now compare our average farm dam CH$_4$ emission with the IPCC estimate in the discussion:

"*Average CH$_4$ fluxes from our farm reservoirs correspond to 417 kg CH$_4$ ha$^{-1}$ yr$^{-1}$, which is greater than the current IPCC emission factor estimate of 183 kg CH$_4$ ha$^{-1}$ yr$^{-1}$ (IPCC, 2019). Considering the skewness of our CH$_4$ data, our median value of 184 kg CH$_4$ ha$^{-1}$ yr$^{-1}$ agrees with the emission factor of other artificial ponds.*" Line 368

L36. It's worth noting the recent paper by van Bergen et al who measured CH4 (including ebullition) and CO2 emissions, and C burial of an urban pond. Ideally we need studies that quantify GHG emissions and C burial, so the net balance can be calculated. van Bergen, T.J., Barros, N., Mendonça, R., Aben, R.C., Althuizen, I.H., Huszar, V., Lamers, L.P., Lürling, M., Roland, F. and Kosten, S.,

2019. Seasonal and diel variation in greenhouse gas emissions from an urban pond and its major drivers. Limnology and Oceanography.

**Response**: The van Bergen reference has now been added to the following sentences in the introduction.

*"Artificial reservoirs have the potential to be potent sources of $CO_2$ and $CH_4$ (Downing et al., 2008; Holgerson and Raymond, 2016). This can be demonstrated by a carbon budget estimate from an urban pond where carbon emissions (both diffusive and ebullitive for $CH_4$) offset carbon burial by >1,000% (van Bergen et al., 2019)."* Line 59

L60. "Currently, only three studies have comprehensively assessed C fluxes from small agricultural reservoirs." What does "comprehensively" mean in this case? These three studies are slightly different – Ollivier et al did not measure ebullition whilst the other two studies did. Ollivier et al and Paneer Selvam et al were 'snapshot' studies whilst Grinham included some temporally repeated measurements (but didn't measure CO2). So are they all comprehensive really? I accept this is a minor point of language but it does matter. Additionally, there are two other papers that have measured farm ponds. Stadmark et al made repeated measurements of CH4 and CO2 emissions from agricultural ponds created to retain N: Stadmark, J. and Leonardson, L., 2005. Emissions of greenhouse gases from ponds constructed for nitrogen removal. Ecological Engineering, 25(5), pp.542-551. There is also data in an old and rather blandly titled paper from two farm ponds. Baker-Blocker, A., Donahue, T.M. and Mancy, K.H., 1977. Methane flux from wetlands areas. Tellus, 29(3), pp.245-250. L62. "Large fractions of CH4 being released." Fractions seems like an odd and unsuitable word. Change for "volumes", "amounts", "quantities", etc?

**Response**: We have removed "comprehensively" and replaced with "at regional scales" in the sentence which now reads:

*"Currently, only three studies have assessed C fluxes from small agricultural reservoirs at regional scales and these support the notion that they are important landscape sources of GHGs (Panneer Selvam et al., 2014; Grinham et al., 2018a; Ollivier et al., 2019)."* Line 64

Because here we are referring to studies with a high number of sites spanning a regional scale, we will not refer to the other two studies mentioned given they only measured a couple of sites.

L80. The study region occupies a large area, but seeing as temperatures are given it would also be good to give a value (or range) for annual precipitation. Reading on, I see the results says "precipitation ~60% less than the long-term climate average of 390 mm in Regina." Please give the value in the methods.

**Response**: The following sentence has been added to site description:

*"Average annual precipitation in the area ranges from 354 to 432 mm."* Line 89

L86. It says 101 ponds were sampled, but in table 1 some variables have N = 102. Where does 102 come from?

**Response**: We did sample 102 sites but lost GHG measurements from one. Because we are focusing of $CO_2$ and $CH_4$ samples in this study, we will refer to total number of sites as 101 and replace 102 in Table 1.

L113, L118. Floating chambers are not "incubations". This word should be altered to something like "deployments" or similar. L121. It says DO was measured in mg/l but in table 1 it is given as %. The methods text should be amended to % instead.

**Response**: "Incubations" have now been replaced with "deployments". DO units have also been amended to read % saturation in Methods text.

L149. Inflow is mentioned here. Do these systems have inflows? Is water pumped in for storage, or do they simply collect rainwater?

**Response**: With the water isotope mass balance method, inflow here refers to precipitation, snowmelt, and groundwater inputs. These farm reservoirs are designed collect most water than falls on the landscape due to being positioned in depressional area.

L183. "To avoid multicollinearity, correlation coefficients between pairs from Pearson linear correlation tests was used to guide covariate choice before model fitting." This is vague. Did you use a Pearson correlation coefficient of a certain value to decide when multicollinerity was present?

**Response**: Here if the correlation was significant then it was decided that multicollinerity was present. We have added that detail to the sentence, which now reads:

*"To avoid multicollinearity, correlation coefficients and statistical significance (p <0.05) between pairs from Pearson linear correlation tests was used to guide covariate choice before model fitting (Table S1-3)."* Line 199

L197. Something I desperately miss from the paper is a figure allowing the reader to visualise the raw CH4 and CO2 data and its distribution. I strongly advise the addition of a figure to show this. It could take numerous forms, such as a scatter plot of CH4 vs CO2 for all 101 ponds, or a box plot of GHGs (grouped by pond size, or pasture vs cropland), or even a bar plot showing individual concs for 101 ponds (large and unwieldy perhaps, but visually useful). Reading on I see figure.3 has a very small land-use graph, but I think a more obvious, up-front figure would be better.

**Response**: We have now added a figure (Figure 2) to illustrate the distribution of $CO_2$ and $CH_4$ concentrations across all sites. Additionally, we have added Figure S4 and S5 to Supplementary Materials which illustrates scatterplots of all data used in the models.

[Figure]

**Figure 2: Kernel density estimates of CO₂ and CH₄ concentrations measured in 101 farm reservoirs grouped by land use.**

Fig 2 and fig 3. In part this relates to my point above. Wouldn't these figures be improved by adding the underlying data points on to these figures as a scatter? That way the reader can see the model, and the raw data. It would help the reader visually determine the robustness of the models.

**Response**: While we understand where the reviewer is coming from regarding underlying data points, we chose to avoid adding these here as adding raw data to partial effects plots of GAMs does not provide a meaningful way to represent model fit. These figures illustrate the partial effects transformed on the response scale and the fitted relationship between each covariate and the response is affected by all covariates in the model. Instead, we have now provided diagnostic plots in the supplementary material (Figs. S2 and S3) to allow readers to visually assess the robustness of each model. One of these plots shows the observed versus predicted values of our $CO_2$ and $CH_4$ concentrations with the model, where the non-constant variance of the response is visible as increased spread of observations around the 1:1 line (not shown) at higher values of the response.

[Figure]

**Fig. S2: R output of diagnostic plots for carbon dioxide model**

[Figure]

**Fig. S3: R output of diagnostic plots for methane model**

L210. "CO2concentrations displayed a positive response with. . .NOx" Whilst the upper 95% credible interval continues to increase, the black line presumably suggests that CO2 decreases at the highest NOx levels. Is there a mechanism that can explain this? Figure 3 has a land use graph, but figure two doesn't. Even if there is no difference in CO2 between land use a figure would still be interesting to see, and there is room for an extra panel at the bottom right anyway. For the land use panel in figure 3, the categories are pasture, livestock and cropland. However, line 87 in the methods only mentions pasture (n = 80) and cropland (n = 21). Where do these livestock ponds come from?

**Response**: Credible intervals always flair out to some extent as they are the extremes of the data as the estimated smooth function is less-constrained there because there are no additional data beyond the observed range to constrain the fitted function. You would see the same thing in a linear model with a small negative effect (slope), but flaring credible interval. The estimated smooth is that which has highest posterior density and reflects the best estimate given the data of the partial effect of $NO_x$ on $CO_2$; the interval simply reflects the greater uncertainty in the estimate. We have not quantified the probability that the effect is an increasing one here, but given the shape of the upper credible interval, the posterior probability that the smooth effect is increasing is very small, perhaps on the order of a few %. The addition of supplementary figures S4 and S5 shows the distributions and correlations between covariate pairs to demonstrate this.

[Figure]

**Fig. S4: Scatterplot matrices of covariate data used in the CO₂ model showing distribution and correlation pairs**

[Figure]

**Fig S5. Scatterplot matrices of covariate data used in the CH₄ model showing distribution and correlation pairs.**

A plot for $CO_2$ land use model results, Figure 3I, has now been included. We have also corrected the definition of land use types in methods which now mentions livestock:

*"We sampled 101 farm reservoirs between July and August 2017, ranging in surface area from 158 – 13,900 m² (Table 1), including basins in pasture (n = 18), pastures with livestock (n = 62) and cropland (n = 21) sites."* Line 93

[Figure]

**Figure 3: Response patterns farm reservoir $CO_2$ concentrations with abiotic, biotic, hydromorphological, and landscape variables based on GAMs. $CO_2$ was best estimated by a combination of a) DO saturation, b) alkalinity, c) NOx, d) buoyancy frequency, e) interaction between $\delta_I$ and WRT, f) soil CEC, g) and elevation, with soil salinity (h) and land use (I) not significant. Model deviance explained was 66.5%. The response patterns shown are the partial effect splines from the GAM (solid line) and shaded area indicated 95% credible intervals. See Table S4 and Figure S2 for summary of model statistics and model fit with observed data.**

L224. "Our comprehensive spatial analysis revealed wide variations among CO2 and CH4 concentrations between farm reservoirs" As per my previous comment, there's currently no easy way to assess this until the raw data is more visible in a figure.

**Response**: A new figure (Figure 2), has been provided as suggested previously and is now referenced in that text.

L227. "CH4 was most correlated by internal abiotic and biotic mechanisms" Should this not be "most correlated with"?

**Response**: Corrected.

L282. "Additionally, smaller waterbodies with shorter WRT can support higher rates of internal CO2 production due higher rates of allochthonous DOC mineralisation" Needs amending to read "due to"

**Response**: Corrected.

L285. "This mechanism is also suggested by the observation that higher reservoir CO2concentrations are predicted in high CEC soils Alkaline high CEC soils retain more calcium ions within clay particles which releases carbonates and bicarbonates into soil porewater" It seems like something has gone awry in the writing here, and this should be two sentences or some words need removing.

**Response**: Yes, this sentence should be separated into two. This has now been corrected.

L331. "The effect potential effect of sulfate" The first "effect" needs deleting

**Response**: Corrected.

L336. "In contrast to the external drivers found for CO2, local land use had a significant effect on CH4 concentrations in farm reservoirs (Fig.3I), with significantly higher CH4 levels in cropland waterbodies than those in pasture. This finding contrasts with those from Australian farm reservoirs where diffusive CH4 fluxes were 250% higher in reservoirs with livestock compared to crops," I find this section of the discussion interesting. As the authors write, the intensive agricultural practices associated with cropland could be expected to result in elevated CH4 concentrations. Conversely, pasture/livestock emissions would depend on the system (intensive or extensive), livestock, etc. Intensive grassland systems could easily result in high emissions, whilst low-level grazing might result in emissions being less than those from cropland. So cropland > grassland and grassland < cropland are both explicable it seems to me.

**Response**: We agree that for all land use types, the intensity of agricultural production likely governs the effect on methane in the reservoirs, perhaps more so than land use type itself. Although assessing the intensity of each land use is beyond the scope of this research, we have expanded this section of the discussion with mention to livestock intensity:

*"Our finding contrasts with those from Australian farm reservoirs where diffusive $CH_4$ fluxes were 250% higher in reservoirs with livestock compared to crops, although the mechanisms responsible for observed differences were inconclusive (Ollivier et al., 2019). This difference could be the result of the intensity of agricultural production, where farm reservoirs supporting high intensity grazing may also experience high $CH_4$ production as demonstrated by a couple of high $CH_4$ concentrations observed in our livestock pasture reservoirs (Fig. 2). In this case it's likely that $CH_4$ levels are more influenced by nutrient loading from the landscape which stimulates eutrophication (Huttunen et al., 2003), as suggested by the biotic variables in our model (Fig. 4). The intensity of agricultural production under different land use types should be an area of further exploration for external controls on farm reservoir GHG production."* Line 357

Figure 4 and fig. 5. The study by Grinham et al of Australian ponds is referenced in the text but doesn't seem to be included in these figures. Is there any reason their data was left out?

**Response**: The Grinham et al., 2018 study is included in Figure 4 under "Artificial ponds, Australia" for the $CH_4$ fluxes. We now realise this reference is not included in the supplemental table referred to in the figure caption. Reference details to this study is now included in Table S6.

L365. "Here, CH4 fluxes were converted to CO2-efluxes using the sustained-flux global warming potential over 100 years" I am not familiar with this metric, and suggest a few lines are included in the methods as to what it is and how it is calculated.

**Response**: We have added details to how the $CH_4$ fluxes were converted to $CO_2$-equivalent fluxes in the methods:

*"For comparing $CO_2$-equivalent fluxes, $CH_4$ fluxes were converted using the 100-year sustained-flux global warming potential (SGWP, Neubauer and Megonigal, 2015). This metric offers a more attainable measure of ecosystem climatic forcing, assuming gas flux persists over time instead of*

*occurring as a single pulse as quantified using traditional global warming potentials (GWP, Myhre et al., 2013). Here, a SGWP multiplier of 45 was applied to all CH$_4$ fluxes in the literature comparison, which is slightly higher than the traditional GWP of 32 over a 100-year time frame (Myhre et al., 2013)."* Line 130

Section 4.4. What (if any) vegetation colonises these pools? Is there no role for encouraging certain plant species that might promote C uptake? For instance, Moore & Hunt say: "The carbon sequestration assessment of constructed stormwater wetlands and ponds suggests that emergent vegetation is a significant source to the soil carbon pool (compared to allochthonous sources) and a critical component of carbon sequestration in these systems." Moore, T.L. and Hunt, W.F., 2012. Ecosystem service provision by stormwater wetlands and ponds–A means for evaluation?. Water research, 46(20), pp.6811-6823.

**Response**: We agree that vegetation likely plays an important role in sequestering carbon in sediments and have added the following paragraph to the discussion in section 4.4:

*"Studies have also shown the importance of emergent vegetation plant species in sequestering carbon in sediments. Emergent vegetation was found to contribute significantly to the soil carbon pool of stormwater ponds compared to allochthonous sources (Moore and Hunt, 2012). However, in our CH$_4$ model, the significant effect of sediment C:N ratios suggested that an autochthonous organic matter source from either phytoplankton or submerged macrophytes supports greater CH$_4$ production in farm reservoirs. The ability of farm reservoirs to have a negative climate forcing will rely on the balance between GHG fluxes and sediment carbon accumulation. The effect different plant species and other aquatic primary producers have on both these processes needs to be evaluated in future studies as the current design of farm dams within the study area minimises growth of emergent vegetation through steep sides and slopes."* Line 426

L392. "The flux of N2O was constrained in our earlier study (Webb et al., 2019), which found a small CO2-e sink (-89 to -3 mg CO2m-2d-1) for the majority of these farm reservoirs despite high N concentrations." Something of a diversion here, but doesn't this depend on how the data are interpreted though? In your earlier study the median N2O flux was negative, but the mean was positive (with 33% of ponds emitting N2O), whilst in this study (figs 4 and 5) you present mean CH4 and CO2. There's probably a debate to be had concerning what average is most appropriate to use, but note the IPCC Refinement used a mean value calculated from log-transformed values.

**Response**: We thank the reviewer for their insight but have respectfully retained our original presentation. As noted above, we presented both median and mean in the Webb et al. 2019 publication because we wanted to make a clear point that most small agricultural reservoir was, unexpectedly, not a major source of N$_2$O. This result is not highly dependent on the form of the summary statistic (weak sink, weak source; neither are large). Similarly, in this paper, we focus on the mechanisms predicting variation in the C-based GHG fluxes rather than the absolute values. Thus, while we agree that the 'optics' of the presentation (interpretation by readers) of median and mean are slightly different, we feel that this is a 'side issue' better left for the IPCC committees to debate.

End of Referee #1 response

**Anonymous Referee #2**

This paper describes CO2 and CH4 concentration measurements made during the summer season on 101 farm reservoirs in an agricultural region of Saskatchewan, Canada. The authors then use a series of floating chamber measurements to infer diffusive fluxes of these two greenhouse gases at the pond surface via estimations of gas transfer. The authors also collect data on a number of abiotic and biotic landscape/waterbody characteristics that may help predict farm pond GHG concentrations. They then use general additive modeling to describe controls on waterbody concentration. While not currently emphasized, this paper follows up on a previous article that described novel N2O uptake dynamics in these same ponds. The authors emphasize a few findings: 1) more than half of farm ponds are net CO2 sinks, 2) some (19%) farm ponds are net CO2-eq sinks when looking at diffusive emissions, 3) CO2 concentrations are governed most by hydrology/landscape position, 4) CH4 emissions are governed most by autochthonous production.

**Response**: We thank the reviewer for their critical analysis of our study and appreciate suggestions that further link this work to the broader literature. Detailed responses to comments are provided below.

The current framing of this paper is difficult for me to digest given the complete lack of any CH4 ebullition measurements from these systems (and given that fluxes were estimated based on highly uncertain estimates of gas transfer). While the authors acknowledge that their estimates of CO2-eq emissions are likely low due to the lack of ebullition measurements, this is done at the very end of their paper. I think this point should be made sooner as it is an important detail that influences the interpretation of their findings. The relative contribution of ebullition to total methane flux can vary widely from system to system and the controls on the proportion of methane flux that is ebullitive are not well understood (Deemer et al. 2016 BioScience). It would be helpful to know if the authors observed any evidence of ebullition events during their floating chamber surveys? How much ebullition would have to be observed to push the net CO2-eq sink systems towards net-source? Also, what is the uncertainty in sink vs. source estimations due to uncertainty in system gas transfer velocity? To this same end, it is difficult to see the 19% of systems that are net CO2-eq sinks by looking at the authors' figures. Is this because the net CO2-eq sink is very small? For example, Figure 4 does not seem to show that over 50% of the systems in your study were net CO2 sinks. I suggest adding a zero line to your figures and possibly creating an additional figure that shows fluxes site-by-site for the farm ponds in your study. The visual aids currently offered for showing the distribution of your own dataset are sort of overshadowed by a comparison with the broader literature.

**Response:** We agree that ebullition can be a major methane flux pathway and plan on investigating this in future field studies. Because the focus of the study was to assess the mechanistic drivers of $CO_2$ and $CH_4$ concentrations, the survey was designed to optimise data collection from a large number of sites and ebullition measurements were not carried out. Based on your suggestion, we now highlight this detail earlier in the Methods section:

*"To compare with the literature and assess the source/sink behaviour of the reservoirs, diffusive fluxes of carbon dioxide and methane fluxes were estimated for each water body. Given that the focus of the study was to investigate drivers of $CO_2$ and $CH_4$ concentrations across farm reservoirs, ebullition events were not measured during this survey and as such total $CH_4$ fluxes are likely underestimated. Diffusive fluxes were estimated using water column concentrations ($C_{water}$) and average farm reservoir gas transfer velocity ($k_c$) using the following equation:*

$$f_C = k_c(C_{water} - C_{air}), \qquad\qquad (1)"$$
Line 112

We agree that the highly variable nature of gas transfer velocities is the greatest source of uncertainty in flux calculations. As previously mentioned in the manuscript, k600 values for $CO_2$ and $CH_4$ were $1.50 \pm 1.34$ m d$^{-1}$ and $1.64 \pm 1.14$ m d$^{-1}$, respectively. These data, along with the median, range, and calculated $CO_2$ and $CH_4$ fluxes, have now been added to Table 1 (highlighted in bold below) to provide more transparency to the reader. Please also note that flux and k600 data are provided in a GitHub repository (https://github.com/JackieRWebb/Dugouts-CO2-CH4) which will be publicly available upon publication. Finally, we respectfully note that application of uncertainty values for k600 to our fluxes will increase or decrease the sink or source capacity of the systems, but will not change the number of reservoirs that are $CO_2$-eq sinks/sources.

**Table 1: Farm reservoir and landscape physical, hydrological, and chemical characteristics of the study sites (n = 101)**

| | *Units* | *N* | *Mean* | *Median* | *Min* | *Max* |
|---|---|---|---|---|---|---|
| Area | m$^2$ | 101 | 1,312 | 1,040 | 158 | 13,900 |
| Depth | m | 101 | 2.08 | 2.10 | 0.18 | 5.10 |
| Buoyancy frequency | s$^{-2}$ | 99 | 0.01 | 0.005 | 0.00 | 0.03 |
| $\delta^{18}$O inflow | ‰ | 101 | -13.37 | -13.33 | -19.39 | -8.40 |
| Evaporation to inflow | | 101 | 0.46 | 0.43 | 0.04 | 1.58 |
| Water residence time | Years | 100 | 0.76 | 0.66 | 0.08 | 2.51 |
| $CO_2$ | μM | 101 | 42.2 | 14.6 | 1.3 | 326.1 |
| $CH_4$ | μM | 101 | 4.3 | 1.9 | 0.1 | 54.5 |
| **Flux CO₂** | | | | | | |
| *Positive* | **mmol m$^{-2}$ d$^{-1}$** | **47** | **100.1** | **58.1** | **0.1** | **466.2** |
| *Negative* | **mmol m$^{-2}$ d$^{-1}$** | **54** | **-11.9** | **-13.3** | **-21.3** | **-0.1** |
| **Flux CH₄** | **mmol m$^{-2}$ d$^{-1}$** | **101** | **7.1** | **3.2** | **0.4** | **91.5** |
| **k600- CO₂** | **m d$^{-1}$** | **15** | **1.50** | **0.98** | **0.20** | **4.12** |
| **k600- CH₄** | **m d$^{-1}$** | **23** | **1.64** | **1.25** | **0.38** | **4.14** |
| Temperature | °C | 101 | 20.1 | 19.9 | 15.7 | 29.5 |
| Dissolved O$_2$ | % | 101 | 92.6 | 88.9 | 2.3 | 344.0 |
| Salinity | ppt | 101 | 0.9 | 0.5 | 0.1 | 8.6 |
| pH | | 101 | 8.75 | 8.75 | 6.95 | 10.19 |
| Chlorophyll a | μg L$^{-1}$ | 101 | 99.1 | 36.9 | 2.2 | 2,483 |
| NH$_3$ | μg N L$^{-1}$ | 100 | 354.7 | 100.0 | 10.0 | 5,930 |
| NO$_x$ | μg N L$^{-1}$ | 98 | 196.6 | 34.1 | 1.2 | 3,188 |
| TP | μg P L$^{-1}$ | 98 | 285.2 | 80.0 | 8.7 | 6,480 |
| TN | μg N L$^{-1}$ | 98 | 3,082 | 2,360 | 417.5 | 14,280 |
| DOC | mg C L$^{-1}$ | 99 | 31.8 | 29.3 | 4.6 | 90.4 |
| Sediment organic carbon | % | 101 | 5.2 | 3.9 | 0.6 | 31.4 |
| Sediment organic nitrogen | % | 101 | 0.6 | 0.4 | 0.1 | 2.8 |
| Alkalinity | mg L$^{-1}$ | 96 | 245.4 | 219.2 | 71.0 | 755.5 |
| Soil CEC | M-eq 100g$^{-1}$ | 98 | 24 | 24 | 10 | 180 |
| K$_{sat}$ | cm hr$^{-1}$ | 101 | 9.9 | 5.0 | 0.0 | 39.7 |
| Elevation | m | 101 | 627.6 | 598.0 | 484.0 | 997.0 |

As suggested a solid line indicating the threshold between positive and negative fluxes has been added to Figure 5 for better visualisation. The >50% reservoirs that were found to be sinks may be hard to distinguish because our data is highly skewed by some very high concentrations/fluxes. As per the

suggestion of Reviewer 1, this is demonstrated more clearly by the addition of a density plot (Figure 2).

[Figure]

**Figure 5: Range of CO₂ and CH₄ (diffusive) fluxes observed in natural and constructed small (<0.01 km²) waterbodies, including this study (farm reservoirs). Dots represent the mean reported in each study and error bars the range. If no mean value was reported, then the midpoint was inferred as the middle of range (dashed lines). Solid black line distinguished between positive and negative fluxes. All data is from the published literature and references can be found in the Table S6.**

[Figure]

**Figure 2: Kernel density estimates of $CO_2$ and $CH_4$ concentrations measured in 101 farm reservoirs grouped by land use.**

Also, while I am not very familiar with GAMs, I found this analysis a bit opaque and difficult to interpret as currently described. For example, were both N and P variables put into the model and NOx/DIN came out as more important? Also, how were the variables plotted in figures 2 and 3 selected? From what I can gather, you have plotted more than just the variables in the best model. For the sake of discussion, it would be nice to see a consistent set of variables and their relationship to both CH4 and CO2.

**Response**: Variables for each model were selected based on previous knowledge from the literature on the potential mechanisms controlling $CO_2$ or $CH_4$ in freshwater bodies. The model is designed to test the hypothesis of selected environmental controls and included variables representing water chemistry and biology (Table S1), hydrology (Table S2), and external landscape factors (Table S3). As described in the methods, correlation analysis of covariate pairs was first carried out to guide variable selection in the final models as a) some variables represent the same mechanism and are highly correlated (e.g. total N and total P) and b) provided a first assessment of what variables correlated strongest with the response variable within each group of environmental factors. Results of these correlation analysis is provided in Supplementary materials (Table S1-S3). Finally, all variables plotted in Figs 3 and 4 represent those that were included in the GAM and therefore need to be presented, even if some variables came out as non-significant. This reflects modelling best-practice; were we to remove non-significant covariates we would be implying & assuming that the effect(s) on the response were exactly equal to zero, and yet given our data we do not estimate zero effects for these covariates. The model summary statistics and credible intervals on estimated smooth functions or parametric effects presented in the paper include the additional uncertainty that arises from our ignorance of exactly which covariates had the strongest controls on $CO_2$ or $CH_4$. It is from here that we learn what the most important mechanisms are for potentially controlling gas concentrations.

To me, the more novel part of this data set is the high fraction of ponds that are net CO2 sinks. This is also a finding that is most strongly backed by the data that was collected since the conclusion doesn't rely as much on gas transfer estimates and since CO2 ebullition is typically an extremely small fraction of total CO2 emission. The extent of the CO2 sink in these small agricultural ponds could be compared to the lesser extent reported in the global data set of artificial reservoir GHG dynamics (Deemer et al. 2016). It is also interesting that the CO2 sink seems to scale more with landscape and hydrological factors than with ecosystem productivity. While multiple other studies have already emphasized the potential importance of nutrient management/eutrophication on lake, pond, and reservoir methane emissions (see Beaulieu et al. 2019 for a very recent global scale discussion), the findings you present in this paper suggest that landscape placement of farm reservoirs may help buffer GHG emissions independent of trophic status (via carbonate buffering and groundwater DIC chemistry dynamics). See paper by Pacheco et al 2013 in Inland Waters (which asks if eutrophication can reverse the aquatic C budget). To this end, it would also be nice to see plots comparing emission by land use for both CH4 and CO2 (right now the plot is only shown for CH4).

**Response**: We agree and have expanded the following paragraph in discussion to emphasize our findings on $CO_2$ uptake:

*"The negative fluxes observed in our farm dams represents one of the few studied small waterbodies that exhibit $CO_2$ sink behaviour, with most showing net heterotrophy (Fig. 5). Although other studies have noted $CO_2$ sink behaviour in artificial ponds and reservoirs (Peacock et al., 2019; Ollivier et al., 2019), this is the first study to capture such a high proportion (>52%) of CO2 uptake in such systems, with negative fluxes estimated to range between -21 to -0.1 (mean -12) mmol m-2 d-1 for $CO_2$ (Table 1). These flux ranges compare to $CO_2$ uptake of -1 to -11 mmol m-2 d-1 in agricultural eutrophic lakes of North America (Finlay et al., 2010; Pacheco et al., 2013). Studies have shown the importance*

*of eutrophication, leading to net autotrophy, in enhancing $CO_2$ uptake and reversing carbon budgets in lakes (Pacheco et al., 2013). However, a global analysis of GHG fluxes from lakes and reservoirs revealed that the consequence of increased CH4 emissions with increasing trophic status often outweighs the impact of negative $CO_2$ fluxes (Deemer et al., 2016). Here, our model shows the potential importance of reservoir placement within the landscape as a way of reducing $CO_2$ emissions via hydrological and geochemical controls without the added consequence of increased $CH_4$ emissions.”* Line 372

A suggested by yourself and Reviewer 1, land use in now included in Figure 3 for the $CO_2$ model. In addition, the new Figure 2 also shows the raw data distribution for $CO_2$ concentrations by land use.

[Figure]

**Figure 3: Response patterns farm reservoir $CO_2$ concentrations with abiotic, biotic, hydromorphological, and landscape variables based on GAMs. $CO_2$ was best estimated by a combination of a) DO saturation, b) alkalinity, c) NOx, d) buoyancy frequency, e) interaction between $\delta_I$ and WRT, f) soil CEC, g) and elevation, with soil salinity (h) and land use (I) not significant. Model deviance explained was 66.5%. The response patterns shown are the partial effect splines from the GAM (solid line) and shaded area indicated 95% credible intervals. See Table S4 and Figure S2 for summary of model statistics and model fit with observed data.**

The comparison between human-made and natural waterbodies is also interesting and novel. I think it would be good to more thoroughly introduce this question/concept (that the systems might fundamentally differ from each other) earlier in the paper and then come back to it in the discussion. A good reference for comparing human-made and natural waterbodies is Hayes et al. 2017 L&O Letters as well as Doubek & Carey 2017 Inland Waters.

**Response**: We agree that human-made and natural waterbodies function differently from each other on a range of ecological scales. However, our discussion of the literature review focuses on $CO_2$ and $CH_4$ fluxes only and to date have revealed few differences between constructed and natural systems, mainly because both systems have highly variable flux rates (Lines 382, 388). Given our focus on $CO_2$ and $CH_4$ fluxes here, we did not want to add overly speculative text on the potential impact of human-made and natural waterbodies.

Line by Line Edits

Line 18: add "surface" before "concentrations"

**Response**: Corrected

Lines 20-21: this is a little misleading since pH was actually a better predictor

**Response**: the term "best" has been removed.

Lines 23-24: state the timescale over which you are calculating CO2-equivalents

**Response**: "100-year radiative forcing" has been added.

Line 26: bringing up depth doesn't seem appropriate here since depth didn't come out as a significant predictor variable in your models

**Response**: Depth has been removed from this sentence and revised to more accurately reflect our model findings:

*"From our models, we show that the GHG impact of farm reservoirs can be greatly minimised with overall improvements in water quality and consideration to position and hydrology within the land scape."* Line 25

Line 30-31: Holgerson and Raymond 2016 didn't look at ebullition

**Response**: We have now clarified that this reference refers to diffusive fluxes only: *"Current assessments estimate that diffusive $CO_2$ and $CH_4$ emissions from small ponds ($<0.001$ $km^2$) account for 15% and 40% of global emissions from lakes, respectfully (Holgerson and Raymond, 2016)."* Line 30

Line 45-46: Also check out Couto and Olden 2018. . . there aren't really global papers that distinguish surface area of small farm reservoirs/ponds from small hydropower.

**Response:** We have added "artificial reservoirs" to this sentence to be clear that this global estimate does not just refer to farm reservoirs.

Lines 46-47: I suggest listing out numbers of reservoirs by country since the current phrasing is difficult to interpret. Either that or use a word like "collectively" to indicate that 8 million is the sum across multiple countries.

**Response:** "collectively" has been added.

Line 51: What does It mean to create reservoirs at a rate of up to 60% of standing stock? I'm a bit confused by this wording.

**Response:** "standing stock" has been replaced with "existing reservoirs".

Lines 56-57: It is a bit awkward to suggest that eutrophication results in potent CO2 release since autochthonous production actually works to fix CO2 (see Pacheco et al. 2013).

**Response:** The mention to eutrophication has been removed from the sentence.

Lines 76-77: I suggest clarifying: you are identifying drivers of surface water concentration, not total flux. Although these are related, they are not the same thing.

**Response:** "fluxes" have been replaced with "concentrations".

Lines 86-87: How did you select your sites? Randomly?

**Response**: Sites were selected from a database of farm reservoirs collected by a survey of regional landowners, as well as from sites on federal lands. Site selection was refined by ensuring a relatively even spatial distribution across the study area, while also considering ease of access.

Lines 197-202: What were N:P ratios like in these systems?

**Response**: Total N to P ratios (by mass) varied from 1.4 to 126. Readers will be able to refer to all raw data provided in a Github repository ((https://github.com/JackieRWebb/Dugouts-CO2-CH4) which will be made public upon publication.

Results section: I suggest including a summary of the fluxes you estimate (and associated gas transfer rates from the floating chamber surveys). Can you estimate how variability in k might affect variability in your flux estimates? Are there cases where you have both a floating chamber and a concentration based estimate of flux? How much did these differ from each other?

**Response**: As suggested by the reviewer, we have added the summary statistics for both fluxes and measured gas transfer velocities to Table 1. In the results section, we have focused on describing gas concentrations and model results. Instead, description of fluxes are presented later in the paper to aid with comparison of literature values.

Line 227: change "by" to "with"

**Response**: Corrected

Line 246: Not a complete sentence.

**Response**: Sentence corrected to read *"Here, we see evidence for both linked and divergent processes (Fig. 3A)."* Line 261

Lines 261-262: This doesn't seem like a very satisfying explanation to me. Is it also possible that differing hydrology leads to the more stratified systems also being the ones that are higher in CO2?

**Response**: We agree that this sentence is speculative and have removed it.

Line 269: add "of" between "effect" and "increased"

**Response**: Corrected

Line 270: Nitrification doesn't produce CO2; it is an autotrophic process.

**Response**: "nitrification" has been removed.

Line 272: This is a pretty vague topic sentence. It would be helpful to be a little more specific.

**Response**: Sentence has been revised to read: *"Hydrological controls were found to be important regulators of $CO_2$ concentrations in these farm reservoirs."* Line 286

Line 303: get rid of "by"

**Response**: Corrected

Lines 306-307: Deemer et al. 2016 and Beaulieu et al. 2019 are also good references here.

**Response**: References have been added

Lines 312-315: Higher CH4 from higher C:N sediments suggests more (not less) important role for allochthonous C right?

**Response**: Our C/N ratios (8.5 to 13.4) were low enough to still be in the range of autochthonous C based on Liu et al., 2018. However, we have added a sentence to account for the input of

allochthonous C contributing to higher C/N ratios: *"This suggests that in situ rather than terrestrial organic matter (OM) was likely the main source of C fuelling methanogenesis in these reservoirs, although increasing $CH_4$ concentrations with C/N may also represent a larger contribution of terrestrial OM."* Line 328

Line 318-319: I would expect thermal stratification to influence bottom water CH4 concentration more than surface water CH4, but you only have surface water concentrations in your model.

**Response**: Yes, this is most likely the case. We have clarified the sentence to read:

*"Thermal stratification of the water column did not significantly influence surface $CH_4$ concentrations in small farm reservoirs (Fig. 4E)."* Line 333

Line 331: Get rid of second "effect"

**Response**: Corrected

Line 334-335: Avoid using the word "clearly". Also, it would be helpful to show the relationship between CH4 and salinity in your Figure 3 to support this discussion.

**Response**: "Clearly" has been removed from the sentence which now reads: *"Evidently, the biological influence on $CH_4$ concentrations appears less pronounced in these larger, low-flow dams."* Line 349. The inclusion of conductivity in the $CH_4$ model already represents a potential sulfate effect and supports this discussion.

Lines 365-366: State the actual factor that you used here too. Was it 34?

**Response**: At the suggestion of Reviewer 1 for additional information on the calculation of $CO_2$-equivalent emissions, this has now been provided in the Methods:

*"For comparing $CO_2$-equivalent fluxes, $CH_4$ fluxes were converted using the 100-year sustained-flux global warming potential (SGWP, Neubauer and Megonigal, 2015). This metric offers a more attainable measure of ecosystem climatic forcing, assuming gas flux persists over time instead of occurring as a single pulse as quantified using traditional global warming potentials (GWP, Myhre et al., 2013). Here, a SGWP multiplier of 45 was applied to all $CH_4$ fluxes in the literature comparison, which is slightly higher than the traditional GWP of 32 over a 100-year time frame (Myhre et al., 2013)."* Line 129

Lines 392-393: It seems like it would be nice to mention this parallel study earlier in your paper and give it a bit more discussion.

**Response**: We agree and now bring attention to this study in the Introduction:

"This study builds on from our previous research farm reservoir GHG research which found an unexpected nitrous oxide ($N_2O$) sink in 67% of reservoirs (Webb et al., 2019)." Line 72

Lines 378-383: This all seems very speculative. As do lines 400-403.

**Response**: We agree that some of the mechanistic narrative is speculative; however, we also feel that our analysis is robust and that these statements provide promising avenues for further testing of tangible solutions for GHG reduction, both by ourselves and other researchers. Consequently, we have respectfully decided to retain this material, unless the editor feels strongly that it should be removed.

We now clarify the mention of building deeper reservoirs as a way to increase water residence time, which was a parameter in our model found to be related to lower $CO_2$ and $CH_4$ concentrations:

[revised manuscript text omitted]
 in generalized linear models, and instead use information from the current set of data to draw predictions. The more flexible modelling approach is useful for uncovering non-standard relationships between predictor and response variables 
[revised manuscript text omitted]